# Major satellite repeat RNA stabilize heterochromatin retention of Suv39h enzymes by RNA-nucleosome association and RNA:DNA hybrid formation

Oscar Velazquez Camacho[1,2,3], Carmen Galan[1†], Kalina Swist-Rosowska[1,2,3], Reagan Ching[1], Michael Gamalinda[1], Fethullah Karabiber[4], Inti De La Rosa-Velazquez[1‡], Bettina Engist[1], Birgit Koschorz[1], Nicholas Shukeir[1], Megumi Onishi-Seebacher[1], Suzanne van de Nobelen[1§], Thomas Jenuwein[1*]

[1]Max Planck Institute of Immunobiology and Epigenetics, Freiburg, Germany; [2]International Max Planck Research School for Molecular and Cellular Biology, Max Planck Institute of Immunobiology and Epigenetics, Freiburg, Germany; [3]International Max Planck Research School for Molecular and Cellular Biology, Faculty of Biology, University of Freiburg, Freiburg, Germany; [4]Yildiz Technical University, Istanbul, Turkey

*For correspondence: jenuwein@ie-freiburg.mpg.de

Present address: †Inmunologia y Genetica Aplicada SA, Madrid, Spain; ‡National Institute of Medical Sciences and Nutrition Salvador Zubiran, Mexico City, Mexico; §RIKILT, Wageningen, Netherlands

Competing interests: The authors declare that no competing interests exist.

**Abstract** The Suv39h1 and Suv39h2 histone lysine methyltransferases are hallmark enzymes at mammalian heterochromatin. We show here that the mouse Suv39h2 enzyme differs from Suv39h1 by containing an N-terminal basic domain that facilitates retention at mitotic chromatin and provides an additional affinity for major satellite repeat RNA. To analyze an RNA-dependent interaction with chromatin, we purified native nucleosomes from mouse ES cells and detect that Suv39h1 and Suv39h2 exclusively associate with poly-nucleosomes. This association was attenuated upon RNaseH incubation and entirely lost upon RNaseA digestion of native chromatin. Major satellite repeat transcripts remain chromatin-associated and have a secondary structure that favors RNA:DNA hybrid formation. Together, these data reveal an RNA-mediated mechanism for the stable chromatin interaction of the Suv39h KMT and suggest a function for major satellite non-coding RNA in the organization of an RNA-nucleosome scaffold as the underlying structure of mouse heterochromatin.

## Introduction

The classic distinction between euchromatin and heterochromatin is used to define decondensed, gene-rich and transcriptionally active regions of the genome vs. more compacted, gene-poor and transcriptionally silent domains (*Heitz, 1928*; *Huisinga et al., 2006*). However, heterochromatin is not transcriptionally inert (*Huisinga et al., 2006*). Mouse heterochromatin contains large portions of non-coding major satellite repeat DNA elements (*Hörz and Altenburger, 1981*; *Vissel and Choo, 1989*). In early mouse development, transcription from the major satellite repeats (MSR) is required to establish heterochromatin formation in the zygote and the 2 to 4 cell embryonic stages (*Probst et al., 2010*; *Casanova et al., 2013*; *Burton and Torres-Padilla, 2014*). Failure of MSR transcription will not allow heterochromatin establishment and results in abrogation of the development of the early mouse embryo (*Probst et al., 2010*; *Burton and Torres-Padilla, 2014*). In somatic mouse cells, heterochromatic transcription is cell cycle regulated and MSR transcripts transiently increase in the late G1/early S phases and persist during mitosis (*Lu and Gilbert, 2007*). Moreover,

RNaseA treatment of permeabilized mouse embryonic fibroblasts (MEF) weakens heterochromatin and leads to dispersion of key heterochromatic factors, such as heterochromatin protein 1 (HP1) (*Maison et al., 2002*). Collectively, these results suggest that mouse heterochromatin contains an RNA component(s) that is required for the initiation and probably also structural stability of heterochromatin.

Nuclear non-coding RNA have been proposed to facilitate recruitment of chromatin factors by serving as guide RNA (*Tsai et al., 2010*) and/or to constitute a structural component of a distinct chromatin configuration (*Rodríguez-Campos and Azorín, 2007*; *Mondal et al., 2010*). For heterochromatin factors, RNA binding has been shown for HP1 (*Muchardt et al., 2002*; *Maison et al., 2011*), the Suv4-20h KMT (*Bierhoff et al., 2014*) and also for other heterochromatin-related proteins that are involved in X inactivation (*Moindrot and Brockdorff, 2016*) and Polycomb-mediated gene silencing (*Bonasio et al., 2014*). RNA association has been proposed to provide an additional affinity, next to DNA and protein-protein interaction and recognition of histone modifications. Several low-affinity interactions can work together and stabilize chromatin association of components of the Polycomb complexes (*Margueron and Reinberg, 2011*).

Non-coding RNA as a structural chromatin entity is well documented for telomeric heterochromatin (*Schoeftner and Blasco, 2009*), has been implicated to be involved in forming DNA:RNA hybrids that could mediate RNAi-directed heterochromatin formation (*Nakama et al., 2012*) and has also been shown to be important for centromere function to ensure kinetochore formation (*Rošić et al., 2014*). RNA preparations from cytoplasmic, nucleoplasmic and chromatin fractions display distinct distributions of a variety of RNA populations that differ in their content for processed messenger RNA, long non-coding RNA and many heterogeneous nuclear transcripts (*Bhatt et al., 2012*). However, the analysis of repeat-rich RNA in this and other genome-wide expression profiles has largely been ignored or understudied. Recent work has documented that repeat-rich RNA sequences that are present in the rapidly reannealing (CoT-1) fraction of the genome extensively decorate nearly all euchromatic regions of interphase chromosomes (*Hall et al., 2014*). This result exposes a general architectural function for non-coding RNA in chromosome organization that extends previous findings for chromatin association of *Xist* and long intergenic nuclear element (LINE) transcripts during X inactivation (*Hall and Lawrence, 2010*; *Chow et al., 2010*).

The molecular mechanisms of how repeat-rich, non-coding RNA initiate and maintain mammalian heterochromatin remain unclear. Here, we address two major questions and examine first whether the chief enzymes for mouse heterochromatin, the Suv39h KMT, contain an RNA binding affinity for major satellite repeat transcripts. Second, we analyze the molecular properties and secondary structures of major satellite repeat RNA and study their association with mouse heterochromatin. We show that the Suv39h2 KMT contains an N-terminal basic domain that confers preferred binding to single-stranded MSR-repeat RNA in vitro. To characterize the association of Suv39h enzymes with chromatin, we purified native nucleosomes from mouse ES cells by micrococcal nuclease (MNase) digestion and fractionation in sucrose density gradients. The Suv39h KMT exclusively accumulate in the poly-nucleosomal fractions and this association was attenuated upon RNaseH incubation and entirely lost upon RNaseA digestion of the MNase-processed input chromatin. These data reveal an RNA component to be important for the recruitment of the Suv39h KMT and suggest that an RNA-nucleosome scaffold is the physiological template for the stable association of Suv39h enzymes to chromatin. In addition, RNA preparations that were purified from MNase-solubilized chromatin display sensitivity towards RNaseH, when they are examined with MSR-specific DNA probes. We propose a model, in which mouse heterochromatin is composed of a higher order RNA-nucleosome scaffold that contains MSR RNA:DNA hybrids and significant portions of single-stranded MSR-repeat RNA.

## Results

### Identification and characterization of the full-length mouse Suv39h2 protein

The mouse Suv39h enzymes are presented by two genes, *Suv39h1* and *Suv39h2*. The mouse *Suv39h2* gene contains an additional exon in the 5'UTR region (*O'Carroll et al., 2000*) that encodes 81 amino acids and allows for a larger protein. The full-length mouse Suv39h2 protein has not been

characterized. We cloned the full-length mouse Suv39h2 cDNA (Materials and methods). Suv39h2 differs from Suv39h1 by containing an N-terminal basic domain (amino acid position 1–81) giving rise to a predicted gene product of 477 amino acids (*Figure 1A*).

To demonstrate the authentic Suv39h1 and Suv39h2 proteins, we prepared chromatin extracts from mouse ES cells and fibroblasts (iMEFs) and probed them with antibodies against the endogenous proteins. Immunoblotting with a Suv39h1 antibody indicated endogenous Suv39h1 (48 kDa) in wild type but not in *Suv39h* double-null (*Suv39h* dn) cells (*Figure 1B*). By contrast, a Suv39h2 antibody recognizes an endogenous protein of 53 kDa in both ES cells and iMEF, which is not detected in the *Suv39h* dn cells. We also raised an antibody that is specific for an epitope in the basic domain of Suv39h2 (*Figure 1—figure supplement 1*) that also detects a gene product of 53 kDa (*Figure 1B*). We conclude that Suv39h2 (477 aa) is distinct from Suv39h1 (412 aa) and that the Suv39h2 protein contains three conspicuous domains, the N-terminal basic domain, the chromo domain and the catalytically active SET domain.

## Suv39h1 and Suv39h2 can independently re-establish H3K9me3 and silence MSR transcription in interphase mouse ES cells

To examine the functional roles of Suv39h1 or Suv39h2 independently and to dissect a possible contribution of the basic domain of Suv39h2, we re-introduced epitope-tagged (EGFP) full-length Suv39h1 or full-length Suv39h2 into *Suv39h* dn mouse ES cells with plasmid constructs that drive expression under the control of the $\beta$-actin promoter (*Figure 1C*). In addition, we generated a Suv39h2 mutant that lacks the basic domain from threonine 3 to lysine 81 (T3K81) (*Figure 1D*). Whole cell extracts were analyzed from unsynchronized and nocodazole-arrested cells by western blot with an $\alpha$-GFP antibody. Similar expression levels were observed between full-length Suv39h2-EGFP and the Suv39h2-D(T3K81)-EGFP mutant in unsynchronized cell extracts, while the Suv39h2-D(T3K81)-EGFP signal is increased in mitotically enriched extracts. Expression of full-length Suv39h1-EGFP was appreciably higher, both in comparison with the Suv39h2-EGFP products and with regard to endogenous Suv39h1 in wt ES cells (*Figure 1D*). By contrast, the Suv39h2-D(T3K81)-EGFP mutant and full-length Suv39h2-EGFP showed similar or lower expression levels as compared to endogenous Suv39h2 in wt ES cells. Importantly, these data indicate that there is no massive overexpression of the reintroduced Suv39h-EGFP products and that full-length Suv39h2-EGFP is not dominating the Suv39h2-D(T3K81)-EGFP mutant or the full-length Suv39h1-EGFP protein expression levels.

We then analyzed the potential of the different Suv39h-EGFP products to restore heterochromatin. For this, we compared bulk H3K9me3 levels, localization of the Suv39h-EGFP products at DAPI-positive heterochromatic foci and silencing of major satellite repeat sequences. All Suv39h-EGFP products can re-establish H3K9me3 to wild-type levels (*Figure 1D*) and independently localize to heterochromatic foci (*Figure 2A*). To address the silencing function, we analyzed transcriptional repression of major satellite repeats (MSR) by quantitative PCR on total RNA preparations. There is significant derepression of MSR in *Suv39h* dn mouse ES cells (*Martens et al., 2005*). While Suv39h2-EGFP and Suv39h2-D(T3K81)-EGFP can fully restore MSR silencing, Suv39h1-EGFP still allows two-fold higher MSR transcript levels as compared to wild type ES cells (*Figure 2B*), although the H3K9me3 ChIP signal is similar for all three Suv39h-EGFP cell lines (*Figure 2C*).

These data indicate that Suv39h1 and Suv39h2 can independently re-establish H3K9me3 and MSR silencing in interphase mouse ES cells. It should be noted, however, that silencing of another prominent Suv39h target, long intergenic nuclear elements (LINE) (*Bulut-Karslioglu et al., 2014*), was not restored by any of the Suv39h-EGFP products (*Figure 2—figure supplement 1*). Thus, in contrast to MSR, de novo silencing of intact LINE elements may require the synergistic activity of both Suv39h1 and Suv39h2 enzymes.

## The basic domain of Suv39h2 strengthens retention at mitotic heterochromatin

Heterochromatin is not transcriptionally inert and major satellite expression transiently increases at the G1/S phase of the cell cycle (*Lu and Gilbert, 2007*), with MSR transcripts accumulating at pericentric regions of condensing chromosomes at the G2 phase preceding mitosis (*Lu and Gilbert, 2007*; *Bulut-Karslioglu et al., 2012*). In addition, several key components of heterochromatin, such as SUV39H1 (*Aagaard et al., 2000*) and HP1 (*LeRoy et al., 2009*) are subject to post-translational

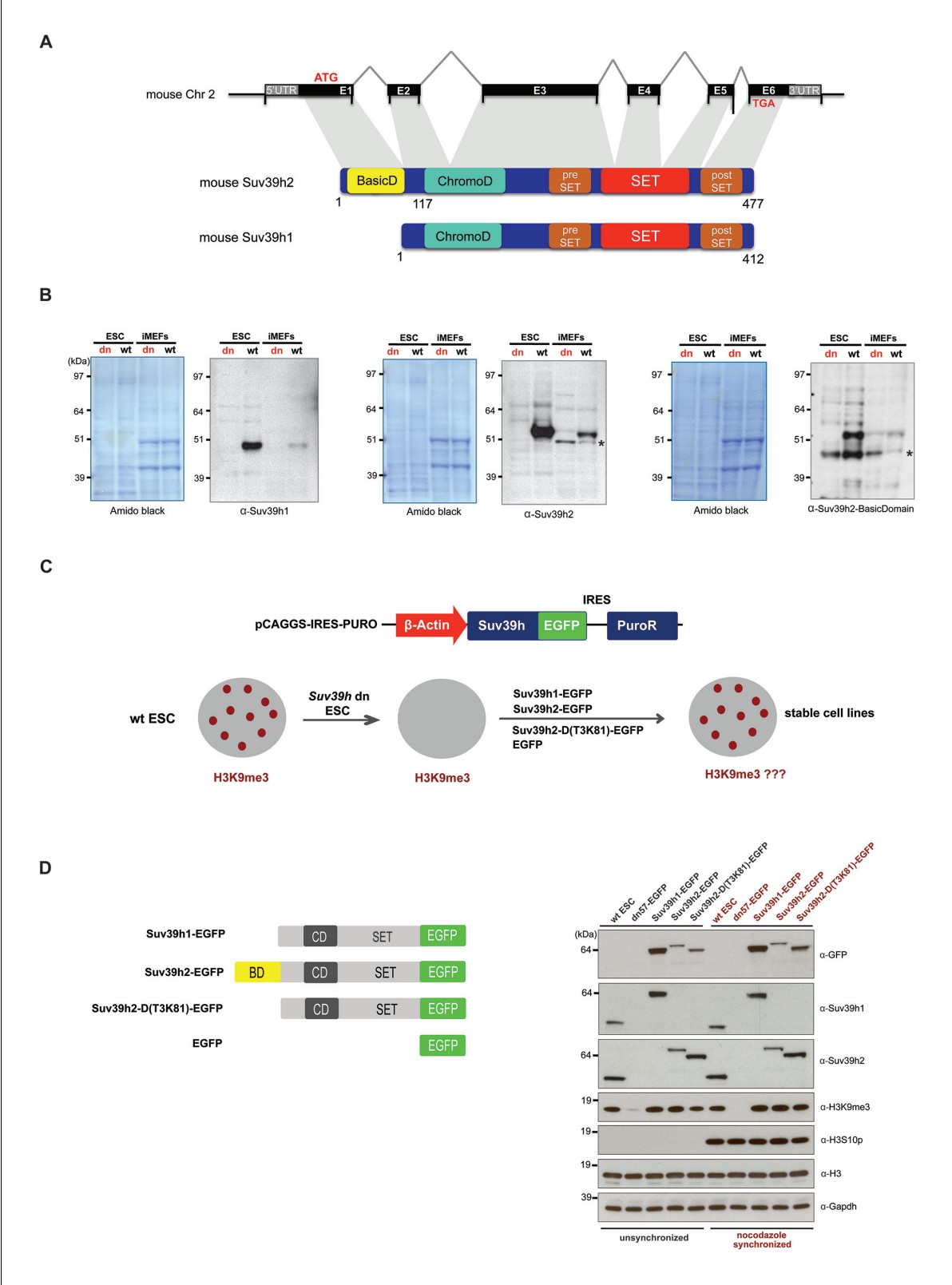

**Figure 1.** Characterization of the Suv39h2 protein and generation of rescued *Suv39h* dn mouse ES cells. (**A**) Schematic representation of the mouse *Suv39h2* gene locus and domain structure of the Suv39h1 and Suv39h2 enzymes showing the N-terminal basic domain of Suv39h2 in yellow. (**B**) Western blot of chromatin extracts from wild type and *Suv39h* dn mouse ES cells (ESC) and fibroblasts (iMEF) to detect endogenous Suv39h1 (48 kDa) and Suv39h2 (53 kDa). An antibody specific for the basic domain of Suv39h2 (**Figure 1—figure supplement 1**) also detects endogenous Suv39h2 at 53 kDa

*Figure 1 continued on next page*

*Figure 1 continued*

in wild type but not in *Suv39h* dn chromatin extracts. The asterisks indicate nonspecific bands. (**C**) Generation of rescued *Suv39h* dn mouse ES cell lines that express the indicated Suv39h-EGFP constructs under the control of a *β*-actin promoter. (**D**) Western blot of whole cell extracts from unsynchronized and nocodazole-synchronized mouse ES cell lines to examine expression of the various EGFP-tagged Suv39h products with an α-GFP antibody or with α-Suv39h1 and α-Suv39h2 antibodies to compare their expression levels with regard to the endogenous Suv39h1 and Suv39h2 proteins. H3K9me3 and H3S10phos levels were also analyzed. Histone H3 and Gapdh served as loading controls.

The following figure supplement is available for figure 1:

**Figure supplement 1.** Generation of polyclonal antibodies against the basic domain of mouse Suv39h2 and amino acid sequence alignment of the basic domain of Suv39h2 with arginine-rich RNA binding factors.

modifications that impair mitotic chromatin association or are released by Aurora-dependent histone phosphorylation (*Fischle et al., 2005*; *Hirota et al., 2005*). To address whether the different Suv39h-EGFP products may have distinct functions at heterochromatin when cells enter mitosis, we used nocodazole synchronization to enrich ES cells at the G2/M phase (Materials and methods). The FACS profile indicates that 89.3% of cells are in the G2/M phase of the cell cycle (*Figure 2I*, below) with around 55% of the nocodazole-arrested cells displaying condensing or mitotic chromosomes (data not shown).

At condensing chromosomes and during pro-metaphase, full-length Suv39h1-EGFP exhibits a broad and diffuse chromosomal pattern (data not shown), but then is largely absent from mitotic chromosomes, although pericentric H3K9me3 persists (*Figure 2D*). In striking contrast, Suv39h2-EGFP displays focal accumulation with the pericentric regions of mitotic chromosomes that is accompanied by higher levels of H3K9me3 (*Figure 2D*). While the Suv39h2-D(T3K81)-EGFP mutant is still enriched at some, but not all, pericentric regions of mitotic chromosomes, it also displays a broader distribution along the chromosomal arms. Larger images of this high resolution microscopy confirm that the Suv39h2-D(T3K81)-EGFP mutant has partially lost the focal accumulation at pericentric heterochromatin of mitotic chromosomes (*Figure 2E*).

In nocodazole-synchronized *Suv39h* dn ES cells, MSR transcripts are similarly derepressed as compared to interphase cells (*Figure 2F*). While both full-length Suv39h1-EGFP and full-length Suv39h2-EGFP products silence MSR expression to levels even lower than those observed in wild type ES cells, the Suv39h2-D(T3K81)-EGFP mutant could only repress transcription to levels near to wild type ES cells (*Figure 2F*). The H3K9me3 ChIP signal at the MSR was also modestly reduced in the Suv39h2-D(T3K81)-EGFP ES cells (*Figure 2G*).

We next wished to analyze chromatin association of the Suv39h-EGFP products with a biochemical assay that allows a better quantification. We prepared whole cell extracts from interphase and nocodazole-synchronized mouse ES cells and incubated them with 10U of micrococcal nuclease (MNase) for increasing time points (Materials and methods). Recovered proteins from the soluble (S) and insoluble (P=pellet) fractions were isolated for further analysis by Western blotting. For this, we included wild type ES cells as a source to compare the endogenous Suv39h1 and Suv39h2 enzymes. Progressive solubilization from chromatin by MNase digestion was quantified by measuring the relative presence of the endogenous or of the Suv39h-EGFP products in the soluble (S) versus the insoluble (P) fraction. A low S/P ratio indicates chromatin retention whereas a high S/P ratio would reveal chromatin release.

In interphase cells, neither endogenous Suv39h1 or Suv39h2 nor full-length Suv39h1-EGFP or full-length Suv39h2-EGFP were significantly released into the supernatant, even after 10 min incubation with MNase (*Figure 2—figure supplement 2*). The Suv39h2-D(T3K81)-EGFP mutant displayed a modestly increased chromatin release.

We then analyzed the MNase mediated chromatin release profile in nocodazole-arrested cells. In pronounced difference to interphase, endogenous Suv39h1 and Suv39h1-EGFP showed rapid solubilization already during the first 4 min of MNase digestion (*Figure 2H*). By contrast, endogenous Suv39h2 and Suv39h2-EGFP largely remained chromatin associated, even after 10 min of MNase digestion. Noticeably, the Suv39h2-D(T3K81)-EGFP mutant was rapidly released into the soluble fraction, with more than 50% of the protein being recovered in the supernatant after 4–6 min of MNase digestion (*Figure 2J*). These results show that Suv39h2 and, to a much lesser degree

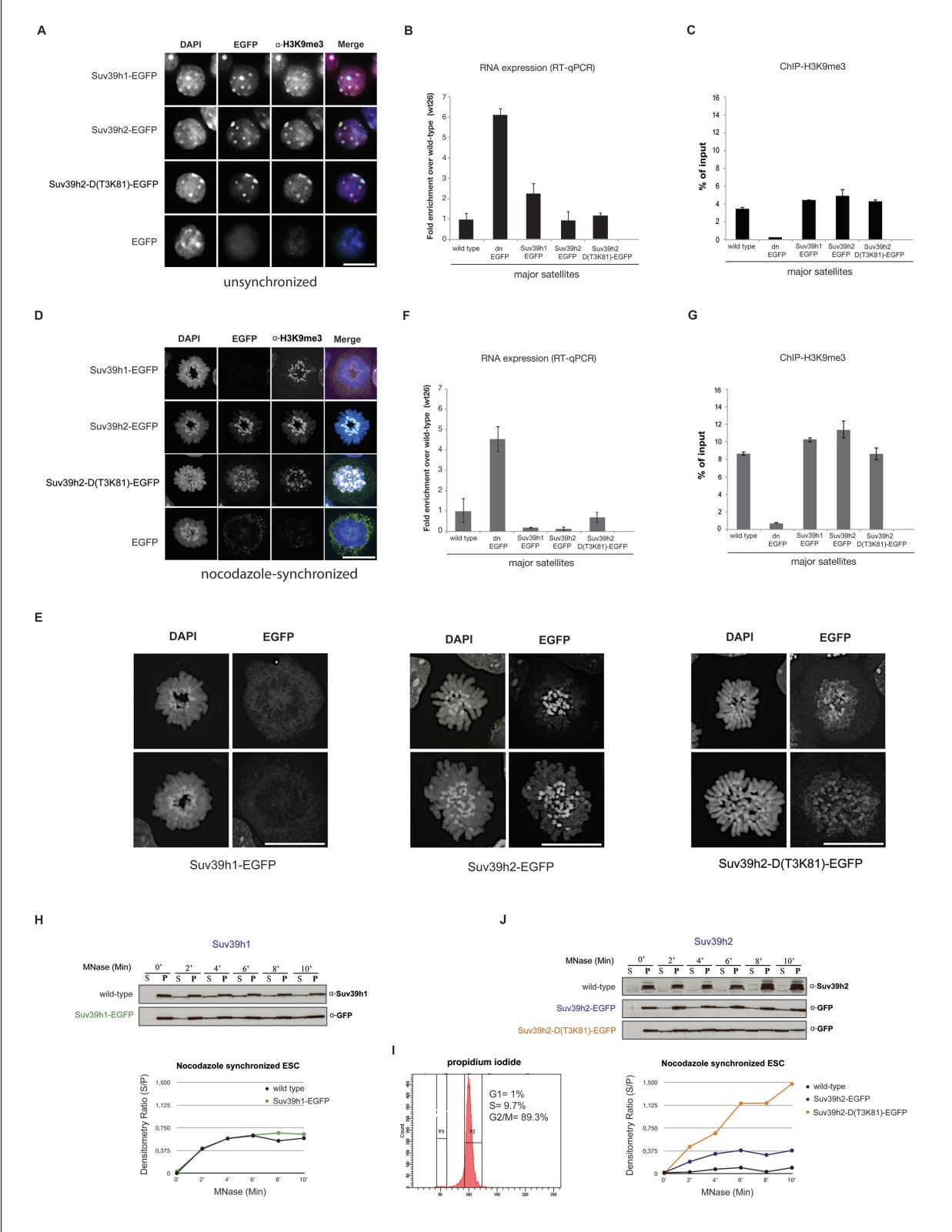

**Figure 2.** Distinct functions of Suv39h1 and Suv39h2 at mitotic chromatin. (**A**) Immunofluorescence analysis for the various Suv39h-EGFP products and for H3K9me3 in interphase nuclei of the rescued mouse ES cell lines. The heterochromatic foci were counterstained with DAPI. Scale bar, 10 μm. (**B**) Reverse transcription quantitative PCR (RT-qPCR) with total RNA isolated from unsynchronized wild type, *Suv39h* dn or Suv39h-EGFP rescued mouse ES cell lines to detect expression from the major satellite repeats with MSR-specific primers. The amplified signals were normalized to *Gapdh* and are

*Figure 2 continued on next page*

*Figure 2 continued*

plotted in the histogram. The data represent the mean ± SD of at least two independent experiments. (C) Directed ChIP for H3K9me3 at the major satellite repeats in unsynchronized wild type, *Suv39h* dn or Suv39h-EGFP rescued mouse ES cell lines. The data represent the mean ± SD of at least two independent experiments. (D) Confocal spinning disc immunofluorescence analysis for the various Suv39h-EGFP products and for H3K9me3 at mitotic chromosomes that were presented in nocodazole-synchronized mouse ES cells. For each image, between 30–50 nuclei displaying mitotic chromosomes were analyzed. Scale bar, 10 µm. (E) Enlarged images of representative confocal IF analyses of mitotic chromosomes as described in (D). Only DAPI and EGFP signals are shown. Scale bar, 10 µm. (F) RT-qPCR as described in (B), but with total RNA preparations from nocodazole-synchronized mouse ES cells. (G) Directed ChIP for H3K9me3 as described in (C), but with chromatin material from nocodazole-synchronized mouse ES cells. (H) Chromatin release assay for endogenous Suv39h1 (wild type) and Suv39h1-EGFP in rescued mouse ES cells. Proteins were detected by Western blot in the soluble (S) or pellet (P) fractions after progressive (0, 2, 4, 6, 8 and 10 min.) MNase digestion of chromatin from nocodazole-synchronized mouse ES cells. Intensity of protein bands in the S or P fraction was quantified by ImageJ software. Chromatin release is measured by the S/P ratio, which is plotted in the indicated graph. (I) FACS profile (propidium iodide labeling) of nocodazole-synchronized wild-type mouse ES cells. (J) Chromatin release assay for endogenous Suv39h2 (wild type), Suv39h2-EGFP and the Suv39h-D(T3K81)-EGFP mutant as described in (H).

The following figure supplements are available for figure 2:

**Figure supplement 1.** Reintroduced Suv39h-EGFP products cannot restore silencing of LINE L1MdA repeats in *Suv39h* dn mouse ES cells.

**Figure supplement 2.** Chromatin release assay in unsynchronized mouse ES cells.

Suv39h1, robustly interacts with mitotic chromatin and resists solubilization by MNase digestion. In addition, the data also indicate a role for the basic domain of Suv39h2 in stabilizing the association of the Suv39h2 enzyme with mitotic chromatin.

## The basic domain of Suv39h2 confers preferred binding to single stranded MSR RNA

The basic domain of Suv39h2 harbors 22 basic residues, of which 19 are arginines (see *Figure 1—figure supplement 1*). Amino acid alignment of this basic domain (aa 1–81) with arginine-rich proteins that were filtered from the SwissProt database reveals modest similarity with RNA binding factors involved in RNA splicing and RNA processing (see *Figure 1—figure supplement 1*). In this respect, the basic domain of Suv39h2 most closely resembles one common RNA binding domain, the arginine-rich motif (ARM) (*Bayer et al., 2005*; *Casu et al., 2013*).

We therefore examined RNA binding of recombinant Suv39h1 and Suv39h2 to MSR repeat RNA in vitro. Recombinant proteins were expressed as 6xHis-MBP-fusions (*Figure 3A*) and probed for binding to one unit (234 nt) of 3'-Cy5-labeled forward (purine-rich) or reverse (pyrimidine-rich) in vitro transcribed (IVT) RNA of the MSR (Materials and methods). Under these conditions and over a protein range from 16 nM to 2 µM, we observed robust band-shift of MSR-F RNA ($K_D$ = 0.8 µM) and of MSR-R RNA ($K_D$ = 0.4 µM) with full-length 6xHis-MBP-Suv39h2 (*Figure 3B*, upper panel). By contrast, full-length 6xHis-MBP-Suv39h1 displayed 3–8 fold weaker binding affinity. Deletion of an extended basic domain (aa 1–116) significantly reduced RNA binding capacity of Suv39h2 (6xHis-MBP-Suv39h2ΔBD) to levels comparable with 6xHis-MBP-Suv39h1, whereas a construct that only expresses this extended basic domain (aa 1–117) of Suv39h2 (6xHis-MBP-Suv39h2basicD(1-117)) maintained high-affinity ($K_D$ = 0.4–0.6 µM) RNA interaction (*Figure 3B*, right panel). As a control, we also probed the full-length 6xHis-MBP-Suv39h1 and Suv39h2 and the 6xHis-MBP-Suv39h2basicD(1-117) product for binding to 3'-Cy5-labeled forward or reverse IVT LINE L1MdA 5'UTR transcripts (208 nt) and did not observe robust RNA interaction (*Figure 3B*, lower panels).

We then detailed these studies and tested for a generic or more specific nucleic acid interaction of the extended basic domain of Suv39h2. For this, we used a GST-Suv39h2basicD(1-117) construct (Materials and methods) and RNA oligonucleotides (35 nt) and complementary DNA oligonucleotides from subunit 2 of the MSR. This allowed us to prepare single-stranded (forward and reverse) or double-stranded RNA or DNA or RNA:DNA hybrids as possible substrates for in vitro binding by GST-Suv39h2basicD(1-117). The data show (*Figure 3C*) that the extended basic domain of Suv39h2 does not bind to ssDNA and dsDNA and there is also no interaction with dsRNA or DNA:RNA hybrids. Instead, GST-Suv39h2basicD(1-117) robustly binds to ssMSR-F and ssMSR-R RNA oligonucleotides.

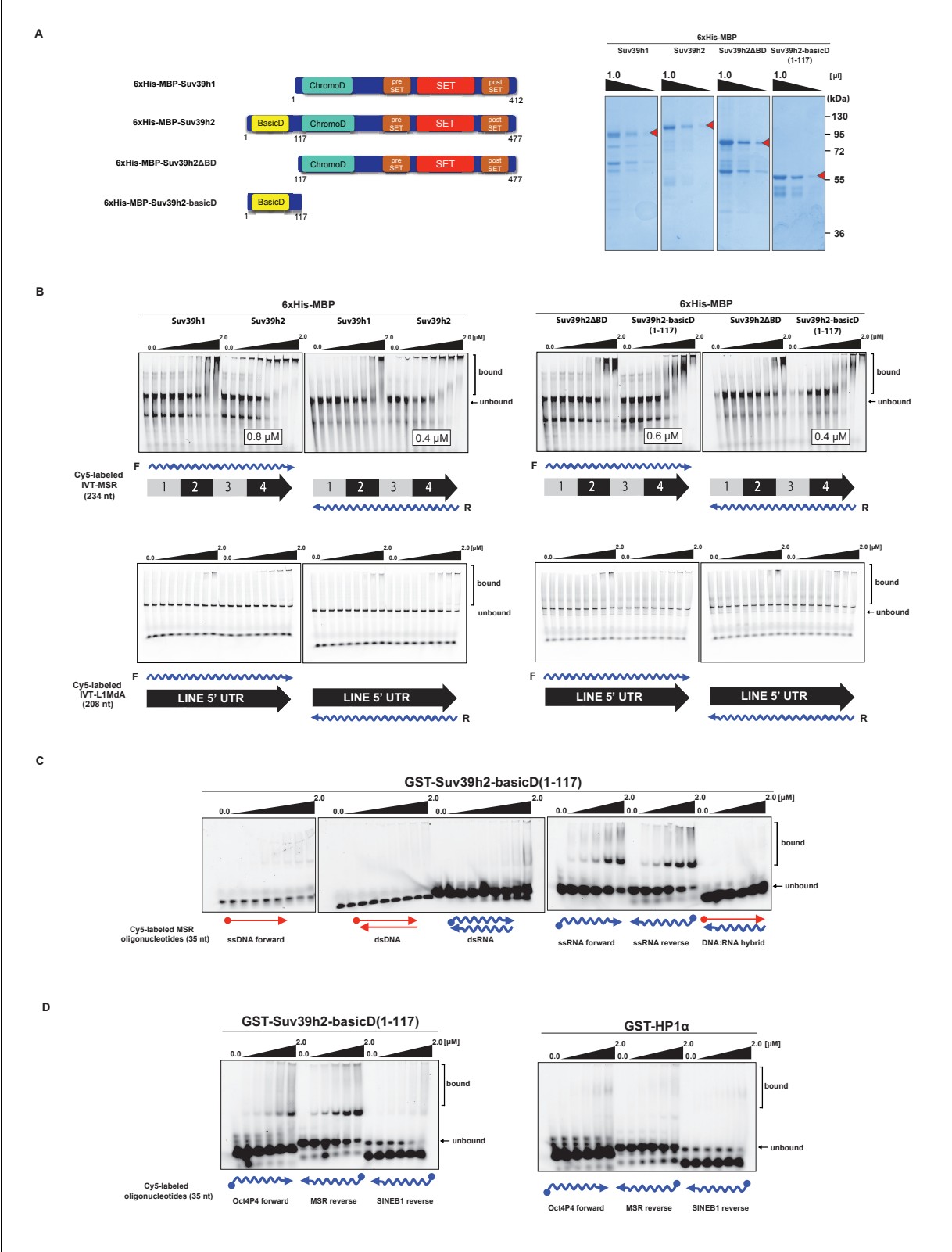

**Figure 3.** The basic domain of Suv39h2 preferably interacts with ssMSR RNA in vitro. (**A**) Schematic representation of full-length Suv39h1, full-length Suv39h2, the Suv39h2ΔBD mutant and the Suv39h2-basicD(1-117) proteins. These were expressed as recombinant 6xHisMBP-fusion proteins and their purity is visualized by Coomassie staining. (**B**) Electrophoretic mobility shift assays (EMSA) with the indicated recombinant 6xHisMBP-Suv39h products and 3'-Cy5-labeled in vitro transcribed full-length (234 nt) ss-forward or ss-reverse MSR transcripts (upper panel) or with 3'-Cy5-labeled in vitro

*Figure 3 continued on next page*

*Figure 3 continued*

transcribed full-length (208 nt) ss-forward or ss-reverse LINE 5'UTR transcripts (lower panel). $K_D$ values that are within the tested protein concentration range of 16 nM to 2 μM were calculated with GraphPad Prism6 software and are indicated in the white boxes. The same amount (50 nM) of IVT MSR or LINE 5'UTR transcripts was used although there was reduced 3'-Cy5 labeling efficiency with the LINE 5'UTR ssRNA. (C) EMSA with recombinant GST-Suv39h2-basicD(1-117) and 5'-Cy5-labeled DNA or RNA oligonucleotides (35 nt each) from subunit 2 of the MSR that are probed as single-stranded, double-stranded or as RNA:DNA hybrid binding substrates. (D) Same EMSA as in (C), but with 5'-Cy5-labeled ssRNA oligonucleotides (35 nt) from an Oct4P4 lnc RNA (*Scarola et al., 2015*), the MSR reverse RNA and a SINE B1 reverse RNA. For comparison, this EMSA was also done with recombinant GST-HP1α.

The following figure supplement is available for figure 3:

**Figure supplement 1.** In vitro RNA binding of GST-Suv39h2-basicD(1-117) towards several distinct RNA oligonucleotides.

To examine selectivity of RNA interaction, we next used RNA oligonucleotides (each at 35 nt) from different repeat classes (minor satellite repeats, LINE L1MdA 5'UTR elements, SINE B1 repeats) and from pRNA of the rRNA cluster (*Schmitz et al., 2010*) or from RNA sequences, such as telomeric TERRA RNA (*Porro et al., 2014*) and a long non-coding Oct4P4-RNA (*Scarola et al., 2015*), that have been shown to bind to SUV39H1. For each of these 12 distinct RNA oligonucleotides, in vitro binding assays with GST-Suv39h2basicD(1-117) over a protein concentration of 16 nM to 2 μM were performed (*Figure 3D* and *Figure 3—figure supplement 1*). GST-Suv39h2basicD(1-117) displayed highest affinity for MSR and TERRA RNA ($K_D \leq 2$ μM), intermediate binding to pRNA, Oct4P4 RNA, minor satellite repeat RNA and LINE L1MdA 5'UTR forward RNA (KD > 2 μM) but did not interact with LINE L1MdA 5'UTR reverse RNA, SINE B1 repeat RNA or with an unstructured poly (A) RNA control (*Figure 3D*, *Figure 4A* and *Figure 3—figure supplement 1*). For comparison, we used GST-HP1α and could not detect robust binding to Oct4P4, MSR or SINE B1 RNA within a 125 nM to 2 μM ligand concentration range (*Figure 3D*). The HP1α related SWI6 protein in *S.pombe* has a described RNA interaction affinity of around 38 μM (*Keller et al., 2012*).

Collectively, these data indicate that the extended basic domain of Suv39h2 provides an ssRNA recognition module that confers preferred selectivity to bind to pericentric (MSR) and telomeric (TERRA) RNA but not to other (e.g. some LINE and SINE B1) RNA (*Figure 3D* and *Figure 4A*) and does not present an unspecific nucleic acid interaction domain. MSR transcript sequences are A/U-rich (64%), whereas the 5'UTR of LINE L1MdA is G/C-rich (61%). In silico modeling for secondary structures predicts several ring-like stretches of single-stranded (unpaired) RNA in the MSR unit, whereas the 5'UTR sequences of the LINE L1MdA are largely folded into dsRNA (data not shown). To validate these predictions, we performed chemical probing of in vitro transcribed MSR-F and MSR-R RNA and of LINE L1MdA 5'UTR-F and LINE L1MdA 5'UTR-R RNA by SHAPE (selective 2'hydroxyl acylation analyzed by primer extension) analysis (*Spitale et al., 2014*) (Materials and methods). SHAPE reagents modify RNA nucleotides in highly flexible or single-stranded regions and are detected as strong stops in primer extension reactions. SHAPE products are resolved by capillary electrophoresis and the SHAPE reactivity can then be used for RNA structural modeling (*Lusvarghi et al., 2013*; *Karabiber et al., 2013*). This SHAPE-directed RNA structural modeling shows that the A/U-rich MSR RNA (for both forward and reverse strands) has extended sections of ssRNA, whereas the LINE L1MdA 5'UTR RNA are primarily folded into dsRNA (*Figure 4B,C*). Intriguingly, the LINE L1MdA 5'UTR-F RNA, where we detect an intermediate binding by the extended Suv39h2 basic domain (*Figure 3—figure supplement 1*), also displays some smaller ring-like regions of unpaired RNA (*Figure 4C*). In silico structural modeling (Materials and methods) of the 12 distinct RNA oligonucleotides reveals that RNA sequences with the highest degree of predicted single-stranded regions (MSR and TERRA) appear as favored substrates for binding by the extended basic domain of Suv39h2, whereas RNA sequences with the lowest degree of predicted single-stranded regions (L1MdA 5'UTR-R and SINE B1) are not bound (*Figure 4D*).

## Suv39h enzymes associate with native nucleosomes in an RNA dependent manner

Based on these in vitro RNA binding data, we next explored whether the association of Suv39h enzymes to native chromatin and nucleosomes may be dependent on an RNA component. For this,

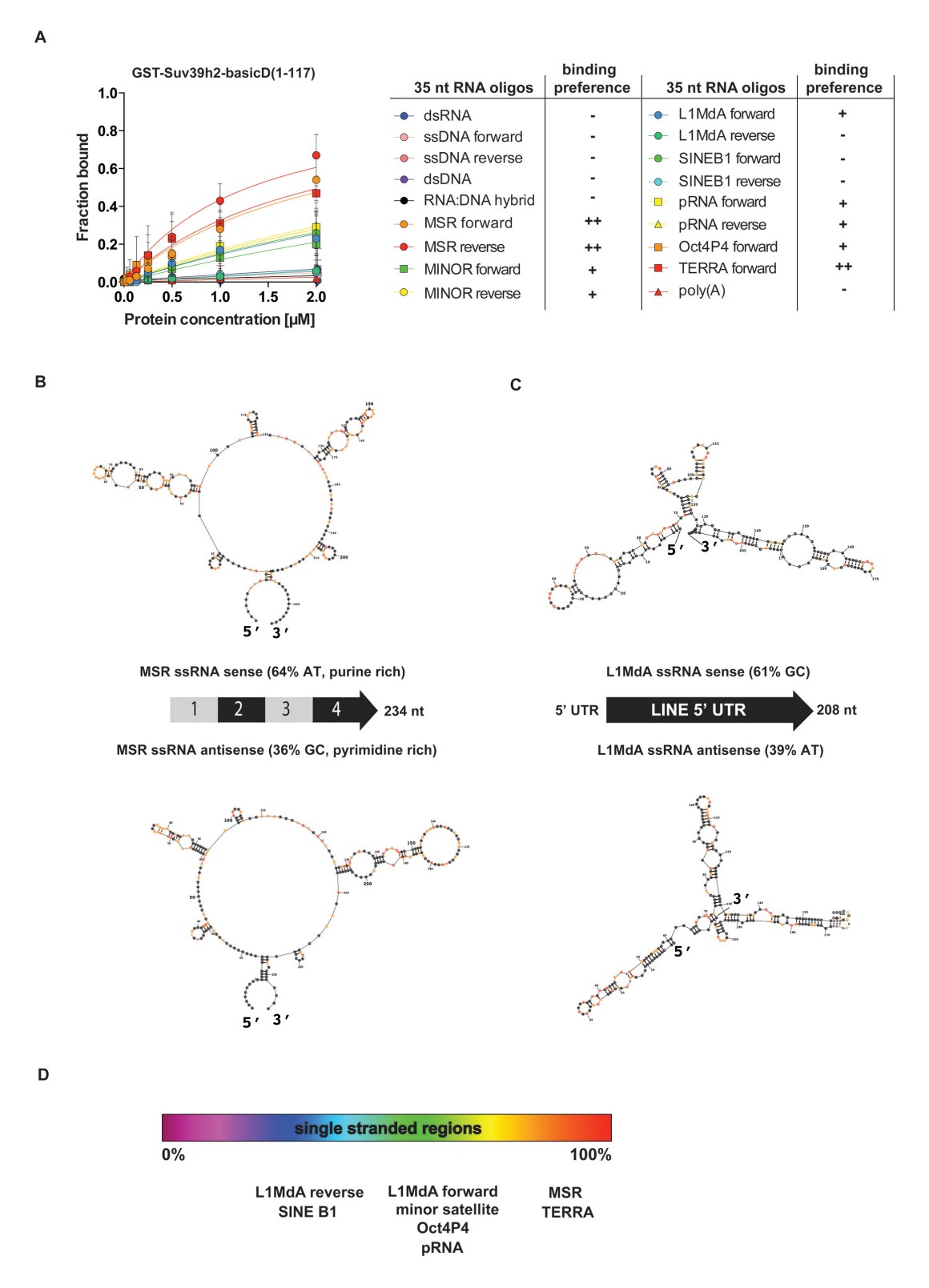

**Figure 4.** SHAPE-directed secondary structure models of major satellite repeat and LINE L1MdA 5'UTR RNA. (**A**) Quantification of in vitro RNA binding for GST-Suv39h2-basicD(1-117) with 12 distinct ssRNA oligonucleotides as shown in *Figure 3C,D* and in *Figure 3—figure supplement 1*. In vitro RNA binding was tested with a protein concentration between 16 nM to 2 μm and is classified as robust binding ($K_D \leq 2$ μM, ++), intermediate binding ($K_D > 2$ μM, +) or no binding (no interaction within this protein concentration range, −). (**B**) Secondary structure models for full-length (234 nt) ss-
*Figure 4 continued on next page*

*Figure 4 continued*

forward (purine-rich) or ss-reverse (pyrimidine-rich) MSR RNA based on in vitro probing by SHAPE analysis. (C) Secondary structure models for full-length (208 nt) ss-forward or ss-reverse LINE L1MdA 5'UTR RNA based on in vitro probing by SHAPE analysis. For both (A) and (B), nucleotide positions of the MSR or LINE L1MdA 5'UTR RNA are indicated and colors reflect normalized SHAPE reactivities of unreactive (0–0.4), moderately reactive (0.4–0.85) and highly reactive (>0.85) nucleotide positions. (D) Classification of the EMSA-tested RNA oligonucleotides (*Figure 3C,D* and *Figure 3—figure supplement 1*) based on their in silico structural prediction of single-stranded regions.

we first isolated chromatin by MNase digestion and separated the nucleosomal fragments on a sucrose gradient (Materials and methods). Such an approach has been used to demonstrate nucleosomal interaction of DNMT3A (*Jeong et al., 2009*) and of other chromatin factors (*Kfir et al., 2015*; *Postnikov et al., 2012*). Purified mouse ES cell nuclei were subjected to partial digestion with 50U of MNase, which cuts linker DNA to generate nucleosomal fragments. The soluble nucleosomal fragments were then either untreated or digested with RNaseH or RNaseA before fractionation on a linear sucrose sedimentation gradient. DNA was extracted from individual fractions and separated on an agarose gel to visualize nucleosome-free (fractions 1–7) and mono- to poly-nucleosomes (fractions 8–17) fractions. In parallel, proteins were recovered, concentrated with affinity columns and processed for Western blotting.

For this nucleosome association analysis, we compared the Suv39h-EGFP products with endogenous Dnmt3a and HP1α. In RNase-untreated samples, all three Suv39h-EGFP products are primarily enriched in poly-nucleosomal fractions and not in the nucleosome-free fractions (*Figure 5A*, first panel). Suv39h2-EGFP and the Suv39h2(T3K81)-EGFP mutant display preference for association with higher-order nucleosomes (fractions 12–17), whereas Suv39h1-EGFP shows a broader fractionation profile that extends into mono-nucleosomes. In agreement with previous findings (*Jeong et al., 2009*), Dnmt3a also accumulates with nucleosomes (fractions 11–15), but much less with the higher-order poly-nucleosomes. By contrast, HP1α sedimented in the nucleosome-free fractions and was only marginally distributed with nucleosomes. These data indicate distinct association profiles of the Suv39h KMT, the Dnmt3a DNMT and HP1α with native nucleosomes (*Figure 5A*, first panel).

In RNaseH-digested samples (*Figure 5A*, second panel), this sedimentation profile was only modestly changed. The nucleosomal ladder could clearly be separated and Dnmt3a accumulates with poly-nucleosomes and HP1α is present in the nucleosome-free fractions. While full-length Suv39h2-EGFP still segregated into poly-nucleosomal fractions 12–15, the Suv39h2(T3K81)-EGFP mutant sedimented in fractions 10–14 and a portion of full-length Suv39h1-EGFP was now also found in nucleosome-free fractions. Although this represents only a slight sedimentation shift, the data can be interpreted to suggest that RNA:DNA hybrids contribute to the interaction of Suv39h enzymes to poly-nucleosomes. In addition, RNaseH treatment resulted in the buildup of the three Suv39h-EGFP constructs and of H3K9me3, but not of Dnmt3a and HP1α, in the pellet (*Figure 5A*, second panel).

We then analyzed RNaseA-treated samples that had been digested under two distinct salt conditions. At 350 mM salt, RNaseA has an attenuated activity that will primarily degrade ssRNA, whereas at 100 mM salt RNaseA will remove ssRNA, dsRNA and the RNA strand of RNA:DNA hybrids (*Figure 5—figure supplement 1*). RNaseA (350 mM salt) digestion of MNase solubilized chromatin resulted in partial dissolution of the nucleosomal ladder and the loss of higher-order nucleosomes in fractions 14–17 (*Figure 5A*, third panel). While the sedimentation of Dnmt3a and HP1α remained largely unaltered, both full-length Suv39h1-EGFP and full-length Suv39h2-EGFP were significantly shifted towards mono- and di-nucleosomes and into the highest molecular weight fraction of the nucleosome-free material. Remarkably, RNaseA (350 mM salt) digestion specifically abrogated chromatin association of the Suv39h2(T3K81)-EGFP mutant, which could no longer be detected in the nucleosomal or nucleosome-free fractions (*Figure 5A*, third panel). These data provide evidence for a function of the basic domain of Suv39h2 in conferring ssRNA-mediated nucleosome association. The RNaseA (350 mM salt) digestion also induced accumulation of all three Suv39h-EGFP constructs, but not of Dnmt3a and HP1α, in the pellet. Finally, we examined RNaseA (100 mM salt) digestion of MNase solubilized chromatin. This low salt condition enhances the activity of RNaseA (*Figure 5—figure supplement 1*) and resulted in a nearly full collapse of the nucleosomal ladder where all but mono- or residual di-nucleosomes can be recovered (*Figure 5A*, fourth panel). Further, none of the Suv39h-EGFP products could be detected in the nucleosomal or the chromatin-free fractions and all

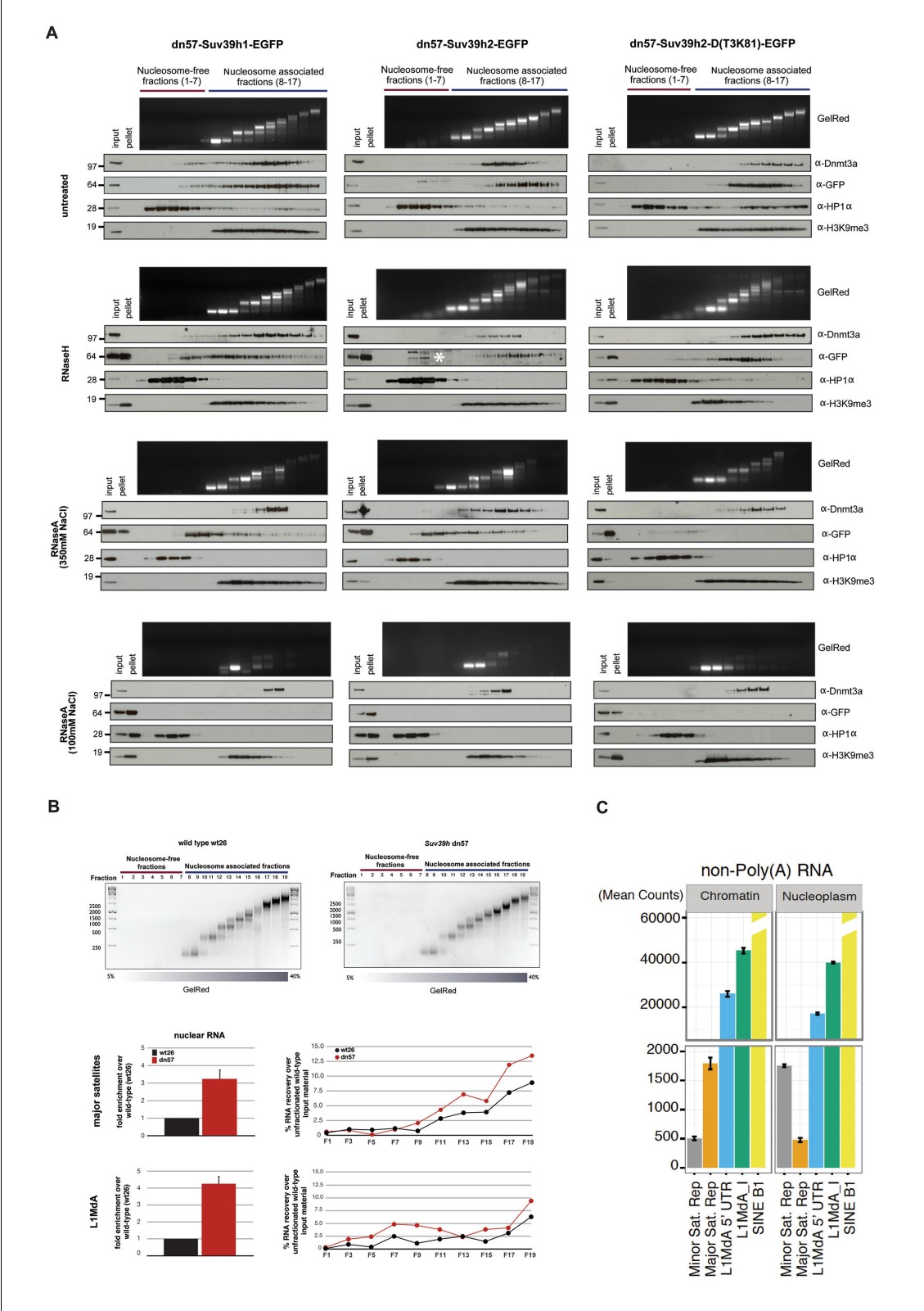

**Figure 5.** RNA-dependent association of Suv39h enzymes to native nucleosomes. (**A**) Sucrose gradient fractionation of MNase-digested chromatin from the Suv39h1-EGFP, Suv39h2-EGFP and Suv39h2-D(T3K81)-EGFP mouse ES cell lines. The separation between nucleosome-free and nucleosome-containing fractions was monitored by Gelred staining of DNA. The sedimentation profile of the various Suv39h-EGFP products and of endogenous Dnmt3a, HP1α and of H3K9me3 was analyzed by western blotting with α-GFP, α-Dnmt3a, α-HP1α and α-H3K9me3 antibodies (first panel). To address

*Figure 5 continued on next page*

*Figure 5 continued*

whether RNA is associated with the nucleosomal fractions, MNase-digested soluble chromatin was incubated with RNaseH (second panel) or with RNaseA at 350 mM salt (third panel) or with RNaseA at 100 mM salt (fourth panel) before being loaded on the sucrose gradient. The asterisk indicates unspecific bands. All of these experiments were performed with two biological replicates and the RNaseA (100 mM salt) and RNaseH treatments were done three independent times. (B) Gelred DNA staining of sucrose gradient fractionation of MNase-digested chromatin from wild type and *Suv39h* dn ES cells and RT-qPCR to detect MSR and LINE L1MdA 5′UTR transcripts in RNA preparations from every second fraction of the sucrose gradient. No signal was detected in the control reactions lacking RT. The histogram on the left shows expression of MSR and LINE L1 MdA transcripts in nuclear RNA of wild-type and *Suv39h* dn ES cells. (C) Hiseq RNA sequencing of chromatin-associated and nucleoplasmic cDNA libraries (non-poly(A) selected) that were generated from wild-type ES cells to quantify the relative abundance of minor satellite repeat, major satellite repeat, LINE L1MdA and SINE B1 transcripts. Plotted are the mean counts of three biological replicates.

The following figure supplements are available for figure 5:

**Figure supplement 1.** In-vitro characterization of substrate specificity of RNaseH and RNaseA.

**Figure supplement 2.** RNaseA-mediated dissociation of Suv39h-EGFP products from DNase1-generated soluble chromatin fractions.

**Figure supplement 3.** Relative abundance of major satellite and other repeat transcripts in cytoplasmic, nucleoplasmic and chromatin fractions.

accumulate in the pellet. HP1α was also significantly shifted to the nucleosome-free fractions 1–3 and became enriched in the pellet. Importantly, not all nucleosome association was lost upon RNaseA (100 mM salt) digestion, since Dnmt3a was still detected in the fractions containing residual amounts of di- to tri-nucleosomes and H3K9me3 (*Figure 5A*, fourth panel).

We confirmed these results with a different approach and generated DNase1-solubilized nucleosomes that again revealed association of the Suv39h-EGFP constructs with higher-order nucleosomes and disruption of this interaction following incubation with RNaseA (100 mM salt), although parts of the nucleosomal ladder persist this RNase treatment (*Figure 5—figure supplement 2*). Together, these data reveal that an RNA component, which appears to be physically bound to nucleosomes, is intrinsically involved in the organization of bulk chromatin. In addition, the data also indicate that an RNA-nucleosome scaffold is the physiological template for the stable association of Suv39h enzymes to chromatin.

## MSR repeat RNA is chromatin associated

We next performed quantitative PCR (RT-qPCR) with MSR- and LINE L1MdA 5′UTR- specific primers on nucleic acid material that was purified from MNase solubilized nucleosomal fractions. As above, nucleosomal fractions were separated on a sucrose gradient (*Figure 5B*) and the associated nucleic acids were purified. These were then digested with TurboDNase and the remaining RNA was converted to cDNA by reverse transcriptase (RT). Reactions with and without RT confirm the detection of RNA sequences. This analysis identified progressive enrichment of MSR RNA in the nucleosome fractions but not in the nucleosome-free fractions and also revealed higher MSR signals with material from *Suv39h* dn ES cells as compared to wild-type ES cells (*Figure 5B*). This enrichment in the nucleosome fractions vs. nucleosome-free fractions was not observed for LINE L1MdA 5′UTR RNA.

Subcellular isolation of RNA populations from cytoplasmic, nucleoplasmic and chromatin fractions further corroborated that around 80% of the total MSR RNA remain chromatin associated (*Figure 5—figure supplement 3*). Although other RNA repeat classes, such as LINE L1MdA and SINE elements are also enriched in the chromatin fraction, the relative ratio of chromatin-bound RNA vs. nucleoplasmic RNA is highest for MSR RNA (*Figure 5—figure supplement 3*). To quantify the abundance of distinct RNA repeat classes in an unbiased manner, we performed Hiseq RNA sequencing of chromatin-associated and nucleoplasmic RNA populations (Materials and methods). High-coverage, non-poly(A) selected cDNA libraries (>80 million reads per sample) were analyzed for the presence of MSR, minor satellite, LINE L1MdA and SINE B1 transcripts. Although MSR reads are around 10–20 fold below the LINE L1MdA reads and significantly less than the more than 100 fold higher amount of SINE B1 reads, it is only the MSR transcripts that give rise to elevated reads in the chromatin-derived (1800 mean counts) as compared to the nucleoplasmic (450 mean counts) material (*Figure 5C*). LINE L1MdA and SINE B1 transcripts show similar mean counts in both chromatin-

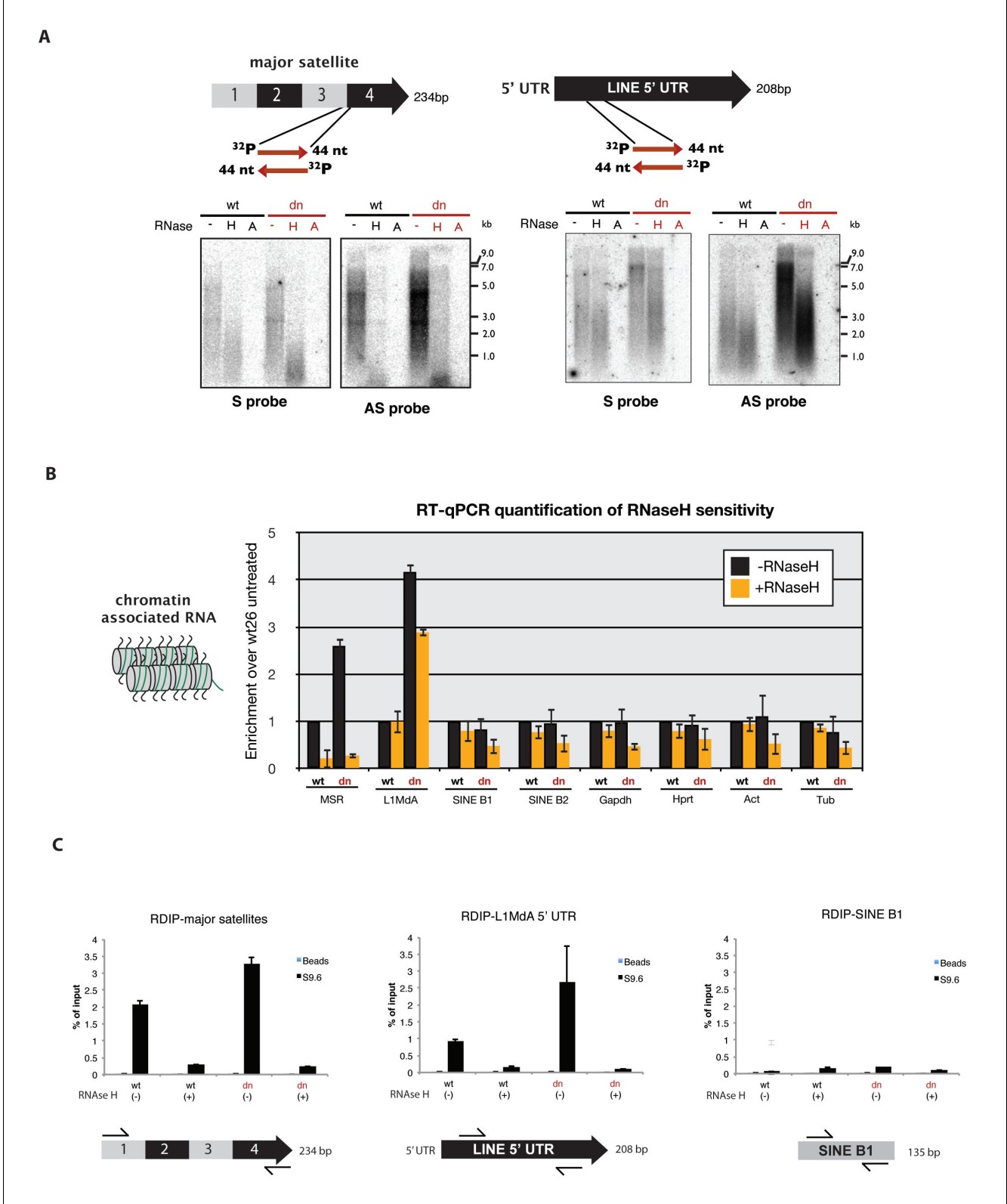

**Figure 6.** RNaseH sensitivity of MSR transcripts and detection of RNA:DNA hybrids. (**A**) Northern blot analysis of chromatin-associated RNA from wild type and *Suv39h* dn mouse ES cells. Equal amounts (5 μg) of trizol-purified chromatin RNA was either left untreated or incubated with RNaseH or RNaseA (100 mM salt) before being separated in a denaturing agarose gel and transferred to a nylon membrane. Hybridization was performed using strand-specific DNA oligonucleotide probes to detect major satellite (left panel) or LINE L1MdA 5′ UTR (right panel) repeat RNA. (**B**) RT-qPCR for
*Figure 6 continued on next page*

*Figure 6 continued*

repeat RNA (MSR, LINE L1MdA, SINE B1, SINE B2) and several housekeeping transcripts (*Gapdh*, *Hprt*, *actin*, *tubulin*) in chromatin-associated RNA isolated from wild type and *Suv39h* dn mouse ES cells and either untreated or digested with RNaseH prior to reverse transcription. The data are normalized to the untreated wild-type control and the experiments were performed with two biological replicates. (**C**) RDIP analysis of chromatin-associated RNA in wild type and *Suv39h* dn mouse ES cells. Trizol-purified chromatin RNA was either untreated or digested with RNaseH and then immunoprecipitated with the S9.6 antibody, followed by directed amplification with primers that are specific for major satellite repeats, the LINE L1MdA 5'UTR or SINE B1 elements. The data represent the mean ± SD of at least two independent experiments.

The following figure supplements are available for figure 6:

**Figure supplement 1.** Characterization of major satellite RNA in mouse ES cells.

**Figure supplement 2.** Specificity of the S9.6 monoclonal antibody towards RNA:DNA hybrids.

derived and nucleoplasmic cDNA libraries, whereas minor satellite repeat RNA is the least abundant in chromatin but enriched in the nucleoplasm. It is important to note that this analysis required cDNA libraries from non-poly(A) selected RNA, as both minor and major satellite transcripts largely lack a poly(A) signal (*Figure 6—figure supplement 1*).

## MSR transcripts are RNaseH sensitive and form RNA:DNA hybrids

Hiseq RNA sequencing data with nuclear RNA preparations from mouse ES cells further show that major satellite repeats are expressed from both strands (*Figure 6—figure supplement 1*) and that the MSR display multiple transcriptional start sites (TSS) (*Bulut-Karslioglu et al., 2012*). Additional analysis of MSR-repeat RNA indicated that it is transcribed by RNA Polymerase II, is not protected at the 5'end and largely lacks a poly(A) tail (*Figure 6—figure supplement 1*). MSR RNA is also chromatin-bound and associates with nucleosomes (*Figure 5B,C*).

To examine the molecular properties of MSR RNA in even more detail, we purified chromatin-associated RNA from mouse ES cells. Strand specific DNA probes for forward (purine-rich) or reverse (pyrimidine-rich) MSR transcripts were used for Northern hybridization with RNA preparations that were either untreated or digested with RNaseH or RNaseA (100 mM salt). In agreement with previous observations (*Lu and Gilbert, 2007*), we detect a heterogeneous population of major satellite transcripts that range from ~0.5 to >7 kb (*Figure 6A*, left panel). Expression of major satellite RNA in the forward orientation (detected by the AS probe) is considerably higher than that for the reverse strand and MSR transcripts are elevated in *Suv39h* dn ES cells. We observe pronounced reduction of hybridization signal for both MSR strands upon RNaseH treatment and complete loss after RNaseA (100 mM salt) digestion (*Figure 6A*, left panel).

We then compared the RNaseH sensitivity of MSR transcripts with those of LINE L1MdA 5'UTR transcripts. LINE L1MdA RNA is also heterogeneous in size and transcribed from both strands. LINE L1MdA RNA is up-regulated in *Suv39h* dn ES cells, but RNaseH digestion only results in a minor reduction of the hybridization signal (*Figure 6A*, right panel). Further, extension of this analysis by using RT-qPCR with chromatin-associated RNA preparations reveals that the RNaseH sensitivity is an intrinsic property of MSR RNA, much less pronounced for LINE L1MdA RNA and not found for SINE B1 transcripts or single copy house-keeping genes (*Figure 6B*).

Finally, we used the monoclonal S9.6 antibody to immunoprecipitate RNA:DNA hybrids from chromatin-associated nucleic acids that were purified from wild type and *Suv39h* dn mouse ES cells. Although the S9.6 antibody is highly specific to detect RNA:DNA hybrids, it also reacts to a minor degree with other non-canonical nucleic acid structures (*Figure 6—figure supplement 2*). The S9.6 antibody has been used in a variety of studies (e.g. *Skourti-Stathaki et al., 2014*; *Nadel et al., 2015*) to detect RNA:DNA hybrids with a procedure, termed RNA:DNA immunoprecipitation (RDIP) (Materials and methods). Chromatin-associated nucleic acids were immunoprecipitated with the S9.6 antibody, followed by directed qPCR with primers that are specific for MSR, LINE L1MdA or SINE B1 elements. Quantification of MSR-RDIP signal indicated 2.1% (wild type ES cells) and 3.2% (*Suv39h* dn ES cells) of the input material to be enriched by the S9.6 antibody (*Figure 6C*). In RNaseH-treated samples, this signal was drastically reduced in both the wild type and *Suv39h* dn ES cell nucleic acid preparations. LINE L1MdA 5'UTR sequences are also enriched, particularly in the *Suv39h* dn ES cell

material, where they too are sensitive to RNaseH digestion. SINE B1 sequences do not appear to be reactive to the S9.6 antibody (*Figure 6C*).

## Discussion

Based on the results presented in this study, we propose an RNA-mediated process to govern the stable association of the Suv39h enzymes at mouse heterochromatin. We also demonstrate that major satellite repeat transcripts remain chromatin-associated and can form RNA:DNA hybrids. Together, these data reveal a novel mechanism for the chromatin association of the Suv39h KMT and suggest a function for major satellite non-coding RNA in the structural organization of mouse heterochromatin. These insights provide evidence for the molecular definition of an RNA-nucleosome scaffold as the underlying structure of mouse heterochromatin.

### MSR RNA govern stable association of Suv39h enzymes with chromatin

Our approach to examine association of Suv39h1 and Suv39h2 with native nucleosomes shows that the Suv39h enzymes are exclusively enriched in the poly-nucleosomal fractions and that this interaction was lost upon RNaseA treatment (see *Figure 5A*). These data suggest an RNA component as an additional mechanism to govern stable chromatin association of the Suv39h KMT and also reveal that several modules in the Suv39h proteins contribute to RNA interaction. While the chromo domain of SUV39H1 has been shown to bind to H3K9me3 (*Wang et al., 2012*), the chromo domains of a number of proteins, for example MSL3 (*Akhtar et al., 2000*) and Chp1 (*Ishida et al., 2012*) can also bind RNA in addition to their interaction with modified histones.

In our study, we identified the extended basic domain of Suv39h2 (aa 1–117) as a novel RNA interaction motif that binds with high affinity to single-stranded transcripts from the MSR RNA in vitro (*Figure 3*). Preliminary mutational analyses suggest that the first 50 amino acids (aa 1–50) of Suv39h2 confer a generic RNA interaction activity, whereas aa 50–117, which have a partial overlap with the very N-terminus of Suv39h1 including three conserved cysteines, determine selectivity of RNA binding (K. S.-R. and TJ, unpublished). The basic domain also protects Suv39h2 from RNaseA-mediated dissociation of native poly-nucleosomes under conditions where RNaseA primarily digests ssRNA, as the Suv39h2-D(T3K81)-EGFP mutant protein is much more sensitive to this RNaseA treatment (*Figure 5A*, third panel). Another function of the basic domain is indicated by strengthening chromatin retention of Suv39h2 with mitotically enriched chromatin (*Figure 2J*) and by ensuring focal accumulation of Suv39h2 with the pericentric regions of mitotic chromosomes (*Figure 2D,E*). Although these data indicate some possible roles of the basic domain to allow for robust heterochromatin accumulation of Suv39h2 during mitosis, we did not detect major defects in heterochromatin association or MSR silencing with the Suv39h2-D(T3K81)-EGFP mutant. Thus, while the basic domain donates an additional RNA affinity, other mechanisms act redundantly to stabilize Suv39h2 association to heterochromatin. Independent work by other groups shows RNA binding activity of the chromo domains of the related mouse Suv39h1 (*Shirai et al., 2017*) and human SUV39H1 (*Johnson et al., 2017*) enzymes.

Together, the cumulative data indicate that the association of the Suv39h KMT to native chromatin is governed by an RNA-mediated process. We suggest that non-coding MSR RNA will organize a higher-order RNA-nucleosome scaffold as the physiological template for the Suv39h enzymes (see below). The RNA dependence of association with poly-nucleosomes is much less pronounced for HP1α and not observed for Dnmt3a (see *Figure 5A*). In fact, HP1α accumulates in the nucleosome-free fractions. Recent studies show that HP1γ, but not HP1α has a preference to interact with nucleosomal H3K9me3 (*Mishima et al., 2015*). Since MSR RNA decorated heterochromatin will provide multiple affinities, such as ssRNA and RNA:DNA hybrids (this study), H3K9me3 (*Wang et al., 2012*), HP1 (*Yamamoto and Sonoda, 2003*; *Maison et al., 2011*), additional chromatin factors (*Maison et al., 2016*) and histone H1 (*Lu et al., 2013*), pericentric accumulation of the Suv39h KMT appears to be maintained by a variety of mechanistic interactions.

These multiple interactions may also explain that we could not observe dispersion of Suv39h1-EGFP or Suv39h2-EGFP from heterochromatic foci in interphase upon RNaseA incubation of permeabilized mouse ES cells, nor define conclusive in vivo interaction of Suv39h enzymes and MSR RNA (rather than MSR DNA) by using PAR-CLIP and RNA-IP. This difficulty may be further complicated by the inherent property of the MSR sequences to form RNA:DNA hybrids. In independent

work, *Johnson et al., 2017* describe dissociation of human SUV39H1 from the pericentric regions of mitotic chromosomes following RNaseA treatment. Human SUV39H1 has been shown to be subject to cell-cycle dependent phoshorylation (*Aagaard et al., 2000*). It is currently unresolved whether mitotic (phosphorylated) SUV39H1 is more sensitive to RNaseA treatment as compared to inter-phase SUV39H1, Suv39h1 or Suv39h2. Also, whether or not Suv39h2 undergoes cell cycle dependent modifications has not yet been analyzed.

## An RNA-nucleosome scaffold as the underlying structure of mouse heterochromatin

Mouse pericentric heterochromatin has long been described to have an RNA component important for the localization of HP1α and for the accumulation of H3K9me3 (*Maison et al., 2002*, *2011*). We show here that MSR transcripts largely remain chromatin-associated (see *Figure 5C* and *Figure 5—figure supplement 3*) and can be detected in poly-nucleosomal fractions (see *Figure 5B*). Moreover, MSR transcripts are sensitive to RNaseH digestion (see *Figure 6A*) and have a secondary structure with extended stretches of unpaired RNA (see *Figure 4B*) that is compatible with the formation of RNA:DNA hybrids. The reiterated major satellite repeats are transcribed from both strands by RNA polymerase II and MSR-repeat RNA does not contain a 5'cap and largely lack a poly(A) tail (*Figure 6—figure supplement 1*). Collectively, these results reveal that heterochromatic MSR-repeat RNA is significantly distinct from a euchromatic messenger RNA that is processed from initiation at a transcriptional start site (TSS) of a dominant gene promoter, to elongation, correct splicing and poly (A)-mediated 3'end formation and then export to the cytoplasm.

RNA-FISH analyses have shown that a sizeable fraction of MSR transcripts remain associated with the pericentric regions of mouse chromosomes (*Lu and Gilbert, 2007*; *Maison et al., 2011*; *Bulut-Karslioglu et al., 2014*; *Ishiuchi et al., 2015*). Although we did not directly address the in vivo stability of MSR transcripts, we propose that these distinct molecular properties of the MSR RNA will induce the formation of a higher-order RNA-nucleosome scaffold that would represent the underlying structure of mouse heterochromatin (*Figure 7*). MSR repeat elements are full of embedded transcription factor binding sites and harbor several TSS that drive transcription of forward (purine-rich) and reverse (pyrimidine-rich) RNA strands (*Bulut-Karslioglu et al., 2012*). Both MSR RNA strands can form RNA:DNA hybrids (see *Figure 6A* and *Figure 6—figure supplement 2*), most likely at the inter-nucleosomal regions. Other non-canonical nucleic acid structures, such as RNA:DNA:DNA triple helices are less likely, although RNA:DNA:DNA triple helices may participate in the regulation of LINE 1 transcription in mouse embryos (*Fadloun et al., 2013*) and in the recruitment of DNMT3b to rDNA loci (*Schmitz et al., 2010*). An RNA:RNA:RNA triple helix has been shown to stabilize the 3'end of lncRNA that lack a poly(A) tail (*Wilusz et al., 2012*). While we have no evidence for direct interaction of Suv39h enzymes or of the basic domain of Suv39h2 with MSR RNA:DNA hybrids or with dsRNA (see *Figure 3C*), several zinc finger transcription factors have been described to interact with RNA:DNA hybrids with affinities comparable or greater than binding to dsDNA (*Shi and Berg, 1995*). The major function of the intrinsic property of the MSR RNA to engage in RNA:DNA hybrid formation may thus reside in the organization of a distinct higher-order nucleosomal structure at pericentric heterochromatin and would be consistent with a proposed role for DNA:RNA hybrids to help directing RNAi-mediated heterochromatin assembly in *S.pombe* (*Nakama et al., 2012*). Additional single-stranded portions of MSR RNA that are not occupied by RNA:DNA hybrid formation will further enforce this higher-order RNA-nucleosome scaffold and provide ssRNA affinity for the basic domain of Suv39h2 or other RNA-binding chromatin factors (*Figure 7*). Recruitment and stabilization of Suv39h enzymes and of additional silencing machineries will then repress or erase (*Keller et al., 2012*) ongoing RNA output at heterochromatin.

Such a transcription-centered model is also in agreement with the proposed functions of heterochromatin to provide RNA quality control (*Zhang et al., 2011*; *Reyes-Turcu et al., 2011*) and RNA surveillance to genomic regions that fail to fully process (*Kowalik et al., 2015*) or export nascent RNA (*Lee et al., 2013*). Stalled splicing (*Dumesic et al., 2013*) and impaired 3'end formation coupled with the aberrant occurrence of R-loops (detected as RNA:DNA hybrids) has been shown to activate an RNAi-like silencing pathway and the induction of G9a-mediated H3K9me2 (*Skourti-Stathaki et al., 2014*). An extension of this mechanism would suggest that the Suv39h enzymes fulfill this role at repeat-rich heterochromatin, where R-loop formation and collisions between RNA transcription and DNA replication appear to be particularly prevalent (*Castel et al., 2014*). While these

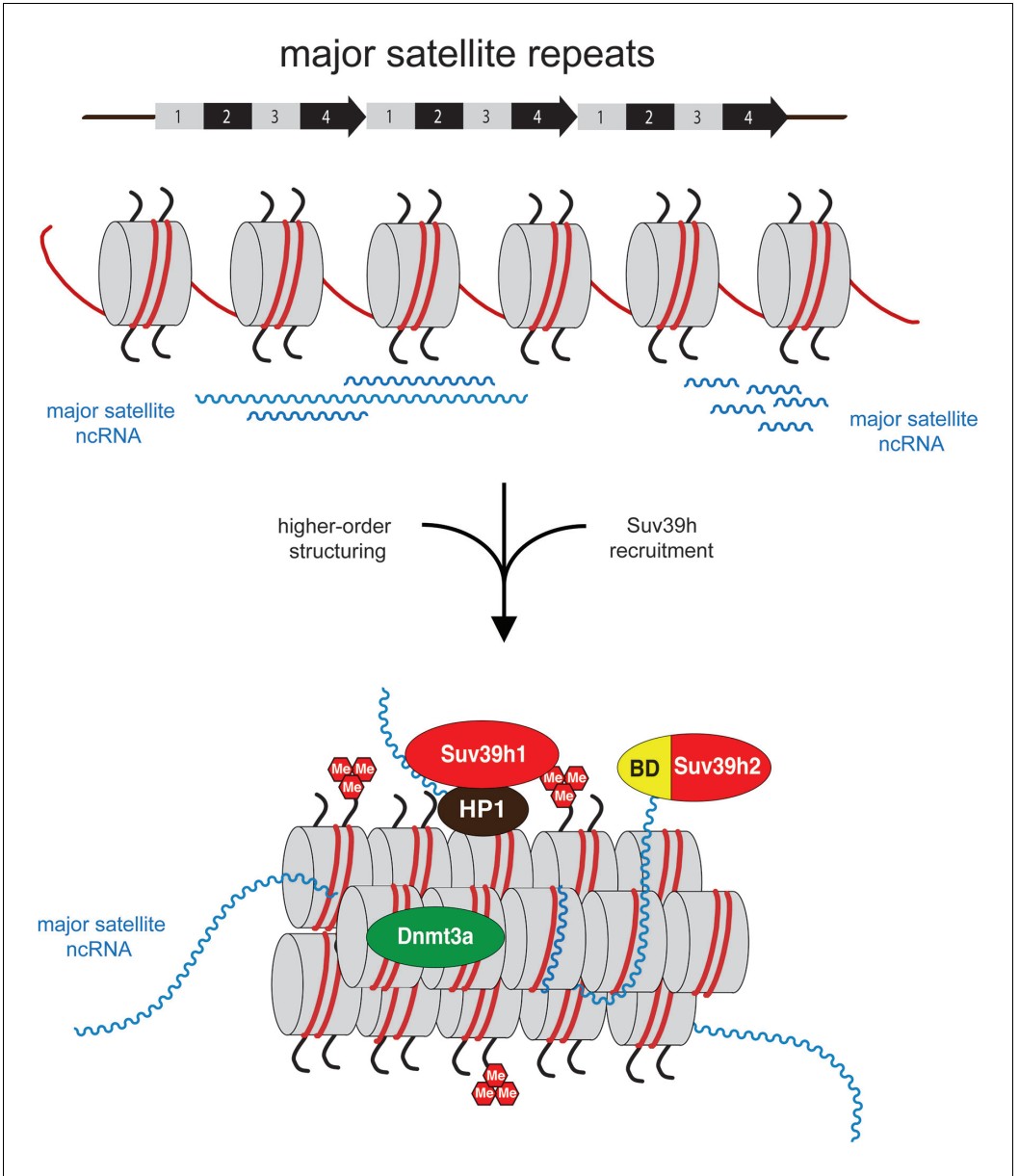

**Figure 7.** Model for an RNA-nucleosome scaffold as the underlying structure of mouse heterochromatin. Model depicting a higher-order RNA-nucleosome scaffold that is established by chromatin association of major satellite repeat (MSR) RNA. In this model, initial transcriptional activity of the MSR repeats is needed to build heterochromatin. The intrinsic property of MSR repeat sequences to form RNA:DNA hybrids will facilitate their chromatin retention and most likely occurs in inter-nucleosomal regions. Additional portions of ssMSR RNA organize the assembly of a higher-order RNA-nucleosome structure and are also important for the recruitment and stabilization of the Suv39h enzymes to heterochromatin. MSR RNA decorated heterochromatin will provide multiple affinities for the Suv39h KMT, such as ssRNA binding by the basic domain (BD) of Suv39h2 (this study), H3K9me3 (**Wang et al., 2012**) and RNA binding by the chromo domains of both mouse Suv39h1 (**Shirai et al., 2017**) or human SUV39H1 (**Johnson et al., 2017**) enzymes and HP1 interaction (**Yamamoto and Sonoda, 2003**; **Maison et al., 2011**). Additional protein-protein contacts with other chromatin-associated components (**Maison et al., 2016**), histone H1 (**Lu et al., 2013**) and transcription factors (**Bulut-Karslioglu et al., 2012**) are not shown.

are intriguing connections, RNA-chromatin association appears to be a much more wide-spread characteristic of mammalian chromosomes, as CoT-1 LINE repeat RNA has recently been shown to decorate most euchromatic regions (*Hall et al., 2014*).

Whereas we discussed MSR-RNA chromatin association for its mechanistic function in directing a distinct heterochromatin conformation, these insights have broader implications for a better understanding of heterochromatin establishment during normal development (*Probst and Almouzni, 2011*; *Burton and Torres-Padilla, 2014*) and its mis-regulation in disease. Aberrant transcription and unscheduled formation of R-loops compromise genome integrity (*Aguilera and García-Muse, 2012*; *Skourti-Stathaki and Proudfoot, 2014*) and accumulate single and double strand DNA breaks, together with an overall increase in chromosome rearrangements (*Huertas and Aguilera, 2003*). Punctate R-loops are also formed when ribonucleotides (rNTP) are mis-incorporated into genomic DNA, causing an increased susceptibility to spontaneous RNA hydrolysis and the formation of DNA nicks in the genome (*Reijns et al., 2012*). Although transient RNA:DNA hybrids have recently been shown to facilitate double-strand break repair (*Ohle et al., 2016*), dys-regulated satellite expression has emerged as a novel tumor marker (*Eymery et al., 2009*; *Ting et al., 2011*; *Zhu et al., 2011*) and RNA:DNA hybrid formation of satellite repeats can drive repeat expansion in tumor cells (*Bersani et al., 2015*).

## Materials and methods

### Molecular cloning of full-length mouse *Suv39h2*

Total RNA from wild type mouse ES cells was converted into cDNA using random hexamers. *Suv39h2*-specific primers (*Supplementary file 1*) complementary to the first and last exon of the *Suv39h2* Ensembl entry ENSMUST00000027956 were then used to amplify the full-length *Suv39h2* cDNA, which was subcloned into the gateway vector pDONR-Zeo. Sequence analysis of the full-length *Suv39h2* cDNA indicates a gene product of 477 amino acids containing a threonine at the third position, consistent with the GenBank accession number AF149205.1.

### Detection of endogenous Suv39h proteins

Chromatin fractions of mouse ES cells and mouse embryonic fibroblasts were prepared as described (*Bhatt et al., 2012*) and processed for Western blotting. For the detection of the endogenous Suv39h proteins, α-Suv39h1 (Cell Signalling D11B6) and α-Suv39h2 (LifeSpan BioSciences LS-C116360) antibodies were applied. In addition, a peptide antibody (Diagenode A1438) that was raised against the basic domain of Suv39h2 was also used (*Figure 1—figure supplement 1*).

### Generation and purification of rabbit polyclonal antibodies against the basic domain of Suv39h2

Suv39h2-basicDomain rabbit polyclonal antibodies were generated by the immunization of rabbits with a double-branched peptide (#C14dbA) comprising amino acids 24 to 39 of the basic N-terminus of Suv39h2 [C-P$_{24}$P RPK ARR TAR RRR AE$_{39}$]2 KC. Crude antisera from rabbit (Diagenode #A1438) was affinity-purified over a HisTrap NHS (GE Healthcare) affinity matrix coupled with 2 mg of the Suv39h2-basicDomain double-branched peptide. Bound antibodies were eluted with 100 mM glycine, NaCl 0.5 M pH 2.7, and further diluted with PBS pH 7.4 supplemented with 0.05% v/v ProClin. The specificity of the affinity-purified antibody (2.0 mg/ml) was confirmed by immunoblotting of Suv39h2 proteins from different sub-cellular fractions of wild type and *Suv39h* double null mouse ES cells.

### Preparation of cytoplasmic, nucleoplasmic and chromatin fractions

The experimental procedure was adapted from *Bhatt et al. (2012)*. Wild-type or *Suv39h* double null mouse ES cells from one confluent maxi dish were trypsinized and collected by centrifugation at 1200 rpm for 3 min in 15 ml falcon tubes. The cell pellet was resuspended in PBS and the cell number determined using a CASY-counter. $2 \times 10^7$ cells were resuspended in 200 μl of cold cytoplasmic lysis buffer (0.15% NP-40, 10 mM Tris pH 7.5, 150 mM NaCl) and incubated on ice for 5 min. The lysate was layered onto 500 μl of cold sucrose buffer (10 mM Tris pH 7.5, 150 mM NaCl, 24% sucrose w/v), and centrifuged in microfuge tubes at 14,000 rpm for 10 min at 4°C. The supernatant

recovered after this spin (~700 µl) represents the cytoplasmic fraction. The nuclear pellet was gently resuspended in 200 µl of cold glycerol buffer (20 mM Tris pH 7.9, 75 mM NaCl, 0.5 mM EDTA, 50% glycerol, 0.85 mM DTT. An additional 200 µl of cold nuclei lysis buffer (20 mM HEPES pH 7.6, 7.5 mM $MgCl_2$, 0.2 mM EDTA, 0.3 M NaCl, 1 M urea, 1% NP-40, 1 mM DTT) was added to the samples, followed by pulsed vortexing and incubation on ice for 1 min. Samples were then centrifuged in microfuge tubes for 2 min at 14,000 rpm at 4°C. The supernatant from this spin represents the soluble nuclear fraction (nucleoplasm), while the remaining insoluble part represents the chromatin pellet. The following antibodies were used for immunoblotting: α-GFP (Invitrogen A11122), α-Gapdh (Sigma G8795), α-H3 (Abcam ab12079), α-H3K9me3 (no. 1926, Jenuwein lab), α-H3S10p (Abcam ab5176).

### Generation of rescued mouse ES cell lines expressing EGFP-tagged Suv39h proteins

Sequences encoding full length Suv39h1, full-length Suv39h2 and the Suv39h2-D(T3K81) mutant were subcloned to generate Suv39h1-EGFP, Suv39h2-EGFP and Suv39h2-D(T3K81)-EGFP fusions and then transferred into the pCAGGS vector that drives expression under the control of the chicken β-actin promoter and confers puromycin resistance. Expression plasmids were transfected into Suv39h double null (Suv39h dn) mouse ES cells using the Xfect reagent (Clontech). Polyclonal cell populations were kept under puromycin selection (1 µg/ml) at all times during subsequent passages. The isolation of wt and Suv39h dn ES cell cultures has been done in the Jenuwein laboratory (*Peters et al., 2003*) and these original cell populations and their derivative cell lines were authenticated by genotyping and Western blot analysis and are routinely tested to confirm that they are mycoplasma free.

### Immunofluorescence analysis and confocal microscopy of mitotic chromosomes

Unsynchronized ES cell cultures expressing the various Suv39h-EGFP products were processed for indirect immunofluorescence to detect EGFP signals and H3K9me3 at heterochromatic foci as described (*Bulut-Karslioglu et al., 2012*). Imaging was performed with an Apotome Axio Z1 (Zeiss) microscope. For enrichment of mitotic chromosomes, ES cell cultures were grown for 6 hr in medium containing nocodazole (Sigma) at a concentration of 0.15 µg/ml. $5 \times 10^4$ ES cells were attached to coated glass slides by using a cytospin (ThermoScientific) at 500 rpm (low acceleration) for 5 min. ES cells were fixed in 4% para-formaldehyde for 15 min at RT, washed three times with 1X PBS and permeabilized in a 0.05% triton-X solution for 5 min. Permeabilized cells were then washed twice with 1X PBS and incubated in blocking solution (PBS/0.25% BSA/0.1% Tween-20% and 10% normal goat serum) for 30 min at RT. The slides were reacted with a α-H3K9me3 rabbit polyclonal antibody (no. 1926, double branched antigen, Jenuwein lab) O/N at 4°C at a 1:1000 dilution. Slides were then washed with 1X PBS and incubated for 1 hr at RT with a secondary goat α-rabbit antibody that was coupled to Cy5 (Life Technologies). Slides were mounted with VECTASHIELD mounting medium containing DAPI. Condensing and mitotic chromosomes presented in the cytospins were examined using a Confocal Spinning Disc microscope (Zeiss Observer Z1) to detect the EGFP signal of the Suv39h-EGFP products and the Cy5 signal for H3K9me3. Images were taken at 63x magnification, analyzed with Zen software 2011 SP3 (Black version) and are shown as 'maximum intensity' projections from Z stacks of representative ES cells (n = 40).

### Chromatin immunoprecipitation (ChIP)

ChIP was performed as described (*Bulut-Karslioglu et al., 2014*) and ChIP-enriched, purified DNA was analyzed by qPCR with primers specific for major satellite repeats, LINE L1MdA 5'UTR and SINE B1 (see *Supplementary file 1*: qPCR primers). For the ChIP detection of H3K9me3, 5 µl crude serum of antibody no. 4861 (Jenuwein lab) per 25 µg chromatin were used.

### Chromatin release assay

Whole cell lysates from unsynchronized and nocodazole-synchronized mouse ES cells were digested with 10 U of micrococcal nuclease (MNase) (Fermentas) for increasing time-points. The reaction was stopped and the samples were centrifuged at 15'000 g for 5 min. Recovered proteins from the

soluble (supernatant) or insoluble (pellet) fractions were processed for Western blotting with the following antibodies: α-GFP (Invitrogen A11122), α-Suv39h1 (Cell Signalling D11B6) and α-Suv39h2 (LifeSpan BioSciences LS-C116360).

## Recombinant protein expression and purification

The 6xHis-MBP-fused full length mouse Suv39h1, full-length mouse Suv39h2, the Suv39h2ΔBD(116-477) mutant and the extended basic domain of Suv39h2 {Suv39h2basicD(1-117)} were obtained as follows. A TEV cleavage site sequence (Integrated DNA Technologies) was cloned into the destination vector p6xHis-MBP using BpII and NotI restriction sites. The p6xHis-MBP-Suv39h1, p6xHis-MBP-Suv39h2, p6xHis-MBP-Suv39h2ΔBD(116-477) and p6xHis-MBP-Suv39h2-basicD(1-117) plasmids were then constructed by cloning PCR products that had been amplified from synthetic Suv39h1 and Suv39h2 coding sequences (Integrated DNA Technologies) into the modified p6xHis-MBP vector and verified by sequencing. 6xHis-MBP fusion proteins were expressed in Rosetta bacterial cells. Cells were grown in LB medium containing 100 µg/ml ampicillin and 0.4% glucose (only required for 6xHis-MBP-Suv39h2), induced with 1.0 mM isopropyl-$\beta$-D-1-thiogalactopyranoside when $OD_{600}$ was 0.8, incubated overnight at 16°C, harvested by centrifugation and lysed in a buffer containing 25 mM Tris-HCl pH 8.0, 350 mM NaCl, 10 mM imidazole, 1% glycerol, 1 mg/ml lysozyme, 0.1 mM PMSF and cOmplete, EDTA-free protease inhibitor cocktail tablets (Roche). Lysate was digested with 25 U/ml Universal nuclease (Thermo Fisher), sonicated (10 times: 10 s ON, 45 s OFF) in a Branson sonicator and centrifuged at 39'000 g for 30 min at 4°C.

Different protocols were then used to optimize purification of the various 6xHis-MBP-Suv39h products. For 6xHis-MBP-Suv39h1 and 6xHis-MBP-Suv39h2ΔBD(116-477), supernatants were affinity purified using 1 ml HisTrap (GE Healthcare) and size purified using Superdex200 (GE Healthcare) columns. The 6xHis-MBP-Suv39h2 product was affinity purified using a 1 ml MBPTrap column (GE Healthcare) followed by cation exchange purification with a MonoS column (GE Healthcare). The 6xHis-MBP-Suv39h2basicD(1-117) was affinity purified using a 1 ml HisTrap column (GE Healthcare) followed by cation exchange purification with a MonoS column (GE Healthcare).

The GST-fused mouse Suv39h2basicD(1-117) was constructed by cloning a PCR product that had been amplified from the synthetic Suv39h2 coding sequence (Integrated DNA Technologies) into the pGEX-6P1 plasmid and verified by sequencing. GST-fused proteins were expressed in BL21 gold bacterial cells. Cells were grown in 2xYT medium containing 100 µg/ml ampicillin, induced with 0.4 mM isopropyl-$\beta$-D-1-thiogalactopyranoside when $OD_{600}$ was 0.8, incubated overnight at 16°C, harvested by centrifugation and lysed in 11 ml (per 0.5 l starting culture) of a buffer containing 40 mM Tris-HCl pH 8.0, 9% glycerol, 2.5 mg/ml lysozyme and cOmplete, EDTA-free protease inhibitor cocktail tablets. Lysate was digested with 25 U/ml benzonase, mixed with 0.5 M KCl, 0.1% NP40, 0.2% Tritonx100, sonicated (20 times: 1 s ON, 2 s OFF) in a Branson sonicator and centrifuged at 12'000 g for 30 min at 4°C. This lysate supernatant (20–30 ml) was affinity-purified by incubation with 0.6 ml Glutathione Sepharose 4B resin (GE Healthcare) at 4°C for 2 hr and 45 min with rotation. The Glutathione Sepharose 4B resin was washed 3 times with 15 ml of buffer containing 40 mM Tris-HCl pH 8.0, 0.5 M KCl, 9% glycerol, 0.1% Tritonx100, 0.1% NP40 and 3 more times with 15 ml of the same buffer supplemented with 0.05 mM $ZnCl_2$. GST-fused proteins were eluted from the washed Glutathione Sepharose 4B resin by incubation with rotation for 30 min at 4°C in 0.6 ml buffer containing 20 mM Tris-HCl pH 8.0, 0.5 M KCl, 9% glycerol, 1 mM DTT and 10 mM reduced glutathione pH 8.0. This elution step was repeated five times and the eluates were combined.

## RNA binding assays

Recombinant proteins for full-length Suv39h1, full-length Suv39h2, the Suv39h2ΔBasicD(116-477) mutant and the extended basic domain of Suv39h2 {(Suv39h2basicD(1-117)} were expressed as 6x-His-MBP-fusions in Rosetta bacterial cells and purified. In addition, the extended basic domain of Suv39h2 and HP1α were also expressed as GST-fusions in BL21 gold bacterial cells and purified. Single-stranded RNA corresponding to either the forward or reverse transcript of one unit (234 nt) of the major satellite repeat (MSR) or of the forward or reverse transcript of one unit (208 nt) of LINE L1MdA 5'UTR (LINE) was generated by in-vitro transcription with T7 RNA polymerase (Thermo Fisher) of PCR products that had been amplified from the pSAT or pEX-L1MdA plasmids using primers containing the T7 promoter sequence. In-vitro transcribed RNA was purified with Qiagen

RNasy Mini Elute Kit followed by 3' end-labeling with a Cy5 fluorophore (Jena Bioscience) using T4 RNA ligase (NEB).

35 nt 5'-Cy5 labeled and HPLC-purified RNA oligonucleotides for MSR, LINE L1MdA 5'UTR, SINE B1, minor satellites, pRNA, Oct4P4, TERRA and poly(A) were purchased from Sigma. For the MSR, also 5'-Cy5 labeled or unlabeled DNA oligonucleotides were purchased (Sigma). To generate dsDNA, dsRNA or RNA:DNA hybrids, equimolar amounts of forward and reverse ssRNA and ssDNA oligonucleotides were mixed in 1xSSC buffer (150 mM NaCl, 15 mM sodium citrate) and incubated for 2 min at 90°C in a Thermomixer. The temperature was decreased to 60°C for 5 min, then reduced to 20°C for 30 min. For EMSA, 50 nM of nucleic acids were mixed with increasing concentrations (16 nM to 2 μM) of recombinant proteins in a buffer containing 20 mM Tris-HCl pH 8.0, 100 mM KCl, 3 mM MgCl$_2$, 1 mM EDTA pH 8.0, 5% glycerol, 0.05% NP40, 2 mM DTT, 50 ng/μl yeast tRNA (Thermo Fisher) and 2.5 ng/μl BSA (NEB). Samples were incubated at 4°C with rotation for 1 hr and then resolved on a 4% polyacrylamide (60:1) gel (25 mM Tris-HCl, 200 mM glycine, 5% glycerol, 0.075% APS, 0.05% TEMED) in 12.5 mM Tris-HCl and 100 mM glycine. The Cy5 signal was scanned on a Typhoon FLA 9500 fluorescence scanner and quantified using ImageJ software. Sequence information for the RNA and DNA oligonucleotides is shown in *Figure 3—figure supplement 1* and *Supplementary file 1* (EMSA).

## RNA secondary structure modeling by in vitro SHAPE

SHAPE-directed RNA secondary structure modeling was performed as described (*Lusvarghi et al., 2013*), with the following modifications. pSAT-S and pSAT-AS plasmids containing one major satellite repeat (234 bp) in either sense or antisense orientation, were linearized with Spe1, gel purified and in vitro transcribed with T7 RNA polymerase (Thermo Fisher). Similarly, single-stranded forward or reverse transcripts of one unit (208 bp) of the LINE L1MdA 5'UTR were generated from pEX-L1MdA-F and pEX-L1MdA-R plasmids. Purified RNA was refolded and treated with 50 mM 2-methyl-nicotinic acid imidazolide (NAI) (Merck Millipore) (*Spitale et al., 2014*) or DMSO (negative control) for 15 min at 37°C. Samples were then reverse transcribed (RT) using distinct fluorophore-labeled primers (*Supplementary file 1*) in RT reactions with NAI-treated RNA (6FAM fluorophore) or untreated RNA (VIC fluorophore). Pooled cDNA synthesis products were then separated by capillary electrophoresis. Electropherograms were processed using the QuSHAPE software package (*Karabiber et al., 2013*). NAI reactivity data were integrated as pseudo-free energy constraints in RNA secondary structure modeling via the RNAstructure (v5.3) software.

In silico structural predictions of RNA oligonucleotides were done with the RNAfold Webserver software (University of Vienna).

## Preparation of native nucleosomes and sucrose gradient centrifugation

Purified nuclei from mouse ES cells were incubated with 50 U of MNase (Fermentas) for 10 min at 25°C. The reaction was stopped with 10 mM EDTA and incubated on ice for 10 min. After centrifugation at 15'000 g for 5 min, the supernatant containing soluble chromatin was either left untreated or incubated with RNaseH or RNaseA. Treatment with RNaseH (Epicenter 10 U/μl) was with 50 U of enzyme to 1 ml of MNase solubilized chromatin for 2 hr at 37°C. Treatment with RNaseA (Fermentas, DNase and protease-free) was with 100 μg of enzyme per 1 ml of MNase-solubilized chromatin for 2 hr at 37°C in a buffer containing either 350 mM or 100 mM NaCl. Samples were then loaded on a linear (5%–40%)%) sucrose density gradient, fractionated by ultra-centrifugation at 36'000 rpm for 14 hr at 4°C and individual fractions (600 μl) were collected. 50 μl aliquots from each fraction was used for DNA analysis. Proteins were concentrated with Amicon Ultra centrifugal filters and processed for Western blotting with the following antibodies: α-GFP (Invitrogen A11122), α-Dnmt3a (Abcam ab23565), α-HP1α (Millipore 05–689-clone15.19s2) and α-H3K9me3 (Abcam ab8898).

## Sucrose gradient fractionation of DNase1-digested chromatin

A nuclear pellet was resuspended in DNase1 digestion buffer (10 mM Tris pH 7.5, 5 mM MgCl$_2$, 0.5 mM CaCl$_2$ and 15 mM NaCl) until a final concentration of 500 ng DNA per μl is obtained. 1 ml of these nuclei was then incubated with 200 U of DNase1 (RNase-Free, Thermo Fisher) for 30 min at 37°C. The reaction was stopped in 10 mM EDTA and incubated on ice for 10 min. After centrifugation at 15'000 g for 5 min, the supernatant containing soluble chromatin was left untreated or

incubated with 100 µg of RNaseA (Fermentas) at low salt for 2 hr at 37°C. Soluble chromatin was then fractionated by ultra-centrifugation in a linear (5%–40%) sucrose density gradient as described above.

## In-vitro characterization of substrate specificity of RNaseH and RNaseA

dsDNA, dsRNA and RNA:DNA hybrids were generated using 35 nt long 5'-Cy5 (for DNA) and 5'-Cy3 (for RNA) labeled oligonucleotides (Sigma) as described above (see RNA binding assays). Single-stranded and double-stranded nucleic acids were incubated with 10 U of RNaseH (Epicenter) in a buffer containing Tris-HCl pH 7.4 and 100 mM NaCl for 30 min at 37°C. RNaseA treatment was carried out with 10 µg of RNaseA (Fermentas) in a buffer containing Tris-HCl pH 7.4 and 350 mM NaCl (high salt) or 100 mM NaCl (low salt) for 30 min at 37°C. Reactions were stopped in a final concentration of 12 nM EDTA, 0.6% SDS and 48% glycerol and incubated on ice for 10 min. Oligonucleotides were diluted to a concentration of 200 nM before separation on a 10% native PAGE gel. Gels were pre-run for 30 min at 150 V at 4°C in 0.5X TBE and the samples were loaded in an equal volume of native loading buffer (30% (v/v) glycerol, 80 mM HEPES-KOH pH 7.9, 100 mM KCl, 2 mM magnesium acetate) and electrophoresed in 0.5X TBE at 150 V using an XCell Sure Lock Midi-Cell Electrophoresis System. After electrophoresis, the gels were scanned in a phosphorimager (GE Healthcare, Typhoon FLA 9500). Sequence information for the 5'- Cy5-DNA and 5'-Cy3-RNA MSR oligonucleotides are listed in *Supplementary file 1* (RNaseA/H activity assay).

## Northern and RT-PCR analysis of chromatin-associated RNA

Subcellular fractionation of mouse ES cells was performed as described (*Bhatt et al., 2012*). Chromatin-associated RNA was extracted with TRIzol from the chromatin pellet and either left untreated or incubated with 50 U of RNaseH (Epicenter, 10 U/µl) or with 5 µl of RNaseA (Thermo Fisher, 20 mg/ml) before separation on a 1.3% formaldehyde agarose gel and processing for Northern blotting. Strand specific DNA oligonucleotide (44 nt) probes (*Supplementary file 1*) were 5' end-labeled with $^{32}$P (Esytides, Perkin Elmer) using T4-PNK and approximately 500,000 c.p.m. of each probe were used for Northern hybridization.

For RT-qPCR detection of chromatin-associated RNA, RNA was extracted with TRIzol from the chromatin pellet. RNA was then incubated with TurboDNase and reverse transcribed (SuperScript II, Invitrogen) using random hexamers. Primers for the specific amplification of repeat elements (MSR, LINE L1MdA, SINE B1, SINE B2) and housekeeping genes (*Gapdh*, *Hprt*, *actin*, *tubulin*) are indicated in *Supplementary file 1*.

## Hiseq RNA sequencing for repeat RNA

RNA was purified from chromatin and nucleoplasmic fractions of wild-type mouse ES cells and converted into non-poly(A) selected, ribosomal RNA depleted (TrueSeq total RNA) or poly(A) selected (TrueSeq mRNA) cDNA libraries following Illumina protocols. The cDNA libraries were sequenced on a HiSeq2500 (Illumina) platform using a 100 bp paired-end approach to give a coverage between 75–88 million reads per cDNA preparation. For each sample, three independent biological replicates were analyzed, resulting in a total number of 268 million reads (chromatin-associated RNA) and 237 million reads (nucleoplasmic RNA) for non-poly(A) selected, ribosomal RNA depleted cDNA libraries or 231 million reads (chromatin-associated) and 226 million reads (nucleoplasmic RNA) for poly(A) selected cDNA libraries. The reads were aligned to the mouse genome build mm10 using TopHat2 (*Kim et al., 2013*) with default parameters and further processed for the detailed analysis of repeat sequences.

## Analysis of repeat element coverage by RNA-seq reads

Using the RNA-seq alignment files, coverage over repeat elements (MSR, minor satellites, LINE L1MdA and SINE B1) present in the RepeatMasker mouse mm10 genome annotation was determined using RepEnrich (*Criscione et al., 2014*), with the following modifications. First, in order to process the RNA-seq alignment files for RepEnrich, uniquely and multiply mapping reads were separated from the alignment file using SAMtools (*Li et al., 2009*) and used as inputs for the RepEnrich tool applying a custom script (RNA_RepEnrich.sh). DESeq2 (*Love et al., 2014*) was then used to obtain normalized counts over repetitive elements present in the modified

RepeatMasker annotation. Second, because RepeatMasker does not separate the 5'UTR from the coding sequence of LINE L1MdA elements, repetitive elements were obtained using the queryRepeatDatabase.pl script from RepeatMasker-4.0.5 (http://www.repeatmasker.org; *Smit et al., 2014*). Sequences corresponding to the 5'end of LINE L1MdA elements, containing both the 5'UTR and ORF1 sequences, were further selected using UCSCtools (*Kuhn et al., 2013*). The 5'UTR of LINE L1MdA elements was then separated from the downstream coding elements by identifying the start of ORF1 using command-line blast tblastn (ncbi-blast-2.2.29+) with ORF1 protein sequence downloaded from Uniprot (UniProt Consortium). The blast hits were further processed using BEDtools (bedtools2-2.25.0). RepeatMasker was then re-run on the mouse mm10 genome using the divided LINE L1MdA sequences (5'UTR, ORF1, and 3'UTR), in order to annotate the 5'UTR of LINE L1MdA elements separately from the downstream ORF1, ORF2 and 3'UTR containing sequences. Mean coverage and SEM of repetitive elements, including GSAT_MM for major satellite repeats, SYNREP_MM for minor satellite repeats, SINE B1_MM and the 5'UTR and downstream sequences of LINE L1MdA elements of three biological replicates were then plotted as barplots using R and ggplot2 (*Wickham, 2009*).

## RNA:DNA immunoprecipitation (RDIP)

A chromatin pellet was prepared from $2 \times 10^7$ mouse ES cell nuclei and sonicated with a Bioruptor (Diagenode) for 5 min using 30 s. sonication cycles. TRIzol purification was then used to extract RNA and associated nucleic acids into the aqueous phase. 7.5 μg of these nucleic acids were either left untreated or incubated with 20 U of RNaseH (Epicenter) for 2 hr at 37°C. Reactions were stopped in lysis buffer and immunoprecipitated with 5 μl of the monoclonal S9.6 antibody (Kerafast) for 3 hr at 4°C and then processed for directed PCR with primers that are specific for MSR, LINE L1MdA and SINE B1 sequences (*Supplementary file 1*).

## RNA polymerase II inhibition

Cultures of wild type mouse ES cells were treated with 100 μg/ml of α-amanitin (Sigma) for 2 hr or 0.1 mM 5,6-dichloro-1-$\beta$-D-ribofuranosylbenzimidazole (DRB, Sigma) for 3 hr to inhibit RNA polymerase II transcription. Total RNA isolated from treated and untreated ESC was digested with the TURBO DNA-free kit (Ambion) and reverse transcribed using random hexamers and reverse transcriptase SuperScript II (Invitrogen). Negative controls were done without including reverse transcriptase during first-strand cDNA synthesis. Specific primers for the amplification of MSR, *Hprt* or 28S rRNA are listed in *Supplementary file 1* (RT-qPCR).

## RNA 5' end analysis

To test for the presence of a 5' cap (m7GpppN), total RNA isolated from wild type ES cells was treated with 5 U of terminator-5'-phosphate dependent exonuclease (Epicenter) for 1 hr at 30°C. This enzyme selectively digests RNA with a free 5'-monophosphate (pN) but cannot digest RNA containing a 5'-triphosphate (pppN), 5' cap (m7GpppN) or 5' hydroxyl group. Following incubation with the terminator exonuclease, the RNA was reverse transcribed into cDNA using random hexamers and the cDNA was further amplified using specific primers for MSR and *Hprt*. Amplification products were separated on a 1% agarose gel.

## Enrichment of poly(A)+ RNA

Total RNA isolated from mouse ES cells was incubated with non-coated beads or with Oligo(dT) coated beads (Thermo Fischer) to enrich for poly(A)+ RNA. Control RNA and poly(A)+ enriched RNA were digested with TurboDNase before being reverse transcribed into cDNA using random hexamers and then amplified using specific primers for MSR and *Hprt*. Amplification products were separated on a 1% agarose gel.

## Characterization of the S9.6 antibody by PAGE and immunoblotting

DNA and RNA oligonucleotides (35 nt) spanning the second sub-repeat of the major satellite consensus sequence were used in the formation of single and double stranded molecules or of RNA:DNA hybrids, as described above (see RNA binding assays). Following separation on 10% native PAGE, gels were transferred to a positively charged nylon membrane (Hybond N+

Amersham) with a semy-dry system. Nucleic acids were then fixed to the membrane with 150 mJ UV light (254 nm) and visualized with SYBRGold (Invitrogen). The membranes were blocked overnight in 0.1% PBS-Tween at room temperature followed by immunoblotting with the S9.6 antibody (Kerafast ENH001) in BSA/PBS overnight at 4°C. After incubation with the secondary goatα-mouse HRP-coupled antibody (Jackson Immuno Research), signal was detected using ECL reagent (GE Healthcare).

## Acknowledgements

We thank Monika Lachner (Freiburg) for help in earlier stages of this work. We are grateful to Xiadong Cheng (Atlanta), Nick Proudfoot (Oxford) and Andrew Jackson (Edinburgh) for insightful discussions and to Maria-Elena Torres-Padilla (Strasbourg) for sharing information on Suv39h2 expression in the early mouse embryo. We acknowledge Andreas Würch (Freiburg) for help with capillary electrophoresis, Kevin Daze and Manoj Rathinaswarmy (Freiburg) for the preparation of full-length recombinant Suv39h proteins, the deep-sequencing unit of the MPI-IE for Hiseq RNA sequencing and Fabian Kilbert from the bioinformatic unit of the MPI-IE for data analysis. Research in the laboratory of T.J. is supported by the Max Planck Society and by additional funds from the German Research Foundation (DFG) within the CRC992 consortium 'MEDEP'. The Marie Curie European fellowship program provided a post-doctoral grant to MG.

## Additional information

### Funding

| Funder | Grant reference number | Author |
|---|---|---|
| Max Planck Institue of Immunobiology and Epigenetics | Open-access funding | Thomas Jenuwein |
| Deutsche Forschungsgemeinschaft | | Thomas Jenuwein |
| The Marie Curie European fellowship program | Postdoctoral Research Fellowship | Michael Gamalinda |

The funders had no role in study design, data collection and interpretation, or the decision to submit the work for publication.

### Author contributions

OVC, Writing—review and editing, Conception and design, Acquisition of data, Analysis and interpretation of data; CG, KS-R, RC, MG, IDLR-V, BE, BK, NS, MO-S, SvdN, Conception and design, Acquisition of data, Analysis and interpretation of data; FK, Supervision, Analysis and interpretation of data; TJ, Writing—review and editing, Conception and design, Analysis and interpretation of data

### Author ORCIDs

Oscar Velazquez Camacho, http://orcid.org/0000-0001-9811-6180
Thomas Jenuwein, http://orcid.org/0000-0002-0470-0421

## Additional files

### Supplementary files

• Supplementary file 1. The Table lists DNA and RNA oligonucleotide sequences that were used as primers or nucleic acid substrates in a variety of assays described in this study. Additional sequences of RNA oligonucleotides (minor satellite repeats, LINE L1 MdA, SINE B1, pRNA and TERRA) used for EMSA are indicated in *Figure 3—figure supplement 1*.

### Major datasets

The following dataset was generated:

| Author(s) | Year | Dataset title | Dataset URL | Database, license, and accessibility information |
|---|---|---|---|---|
| Velazquez Camacho O, Galan C, Swist-Rosowska K, Ching R, Gamalinda M, Karabiber F, De La Rosa-Velazquez I, Engist B, Koschorz B, Shukeir N, Onishi-Seebacher M, van de Nobelen S, Jenuwein T | 2017 | Major satellite repeat RNA stabilize heterochromatin retention of Suv39h enzymes by RNA-nucleosome association and RNA:DNA hybrid formation | http://www.ncbi.nlm.nih.gov/geo/query/acc.cgi?acc=GSE100222 | Publicly available at the NCBI Gene Expression Omnibus (accession no. GSE100222) |

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
