## [Decision Letter]

Thank you for submitting your article "MSR RNA stabilize heterochromatin retention of Suv39h enzymes by RNA-nucleosome association and RNA:DNA hybrid formation" for consideration by *eLife*. Your article has been favorably evaluated by three reviewers, one of whom, Jessica Tyler, also served as Senior and Reviewing Editor. The reviewers have opted to remain anonymous.

The reviewers have discussed the reviews with one another and the Reviewing Editor has drafted this decision to help you prepare a revised submission.

Summary:

The compaction of repetitive DNA elements into heterochromatin is critical for genome stability in eukaryotes. Heterochromatin is molecularly defined by the abundance of histone H3 lysine 9 trimethylation (H3K9me), to which several H3K9me-binding proteins bind, recruiting silencing activities. The Suv39 family of lysine methyltransferases (KMT) catalyze H3K9 methylation in eukaryotes. How Suv39h proteins are recruited to heterochromatin has been the subject of several studies in plants, the fission yeast. and germ cells of flies, worms and mammals. These studies have shown that small and long nuclear noncoding RNAs cooperate to recruit the Suv39 family proteins to repetitive DNA elements, but whether and how nuclear RNAs also play a role in the recruitment of Suv39 proteins in mammalian somatic cells remains unknown. The paper by Camacho et al. addresses this question in mice, demonstrating that the mouse Suv39h2 N-terminal region possesses an RNA binding domain, which can bind to single-stranded long heterochromatic transcripts (such as major satellite repeat (MSR) and TERRA) with high affinity in vitro. They also show that MSR RNAs are tethered to chromatin and that these chromatin-associated RNAs are required for the stable association of Suv39 KMTs to polynucleosomes. These experiments are generally well-controlled and redundantly support each other, and the data are of excellent quality. Unfortunately, these data indicate no functional role for the N-terminal RNA binding domain of Suv39h2 in vivo. However, there is a concern that potential overexpression of the reintroduced Suv39h1 and Suv39h2 may have obliterated the ability to observe a functional requirement for both Suv39h1 and Suv39h2 or to see the functional requirement for the N-terminus of Suv39h2 in vivo.

Essential revisions:

1) How do the expression levels of the reintroduced GFP tagged Suv39h1 and h2 proteins compare to endogenous levels, given that they have more robust silencing and more H3 K9 methylation of major satellites in NZ arrested cells than the wild type (Figure 2)? It would seem likely that the levels may be a lot higher than endogenous levels given that they lead to a much greater silencing of transcription, while the basic domain deletion was reduced to wild type levels. Western blotting with an antibody against Suv39h1 and 2 needs to be included on these westerns. If the levels of the ectopic proteins are a lot higher than endogenous levels, the results need to be discussed and considered in light of this. It would be very unfortunate, because this could explain why either Suv39h1 (which lacks the exon including the N-terminal RNA binding domain) or Suv39h2 can complement the Suv39h1/2 double null mouse cells. This would also explain the ability of Suv39h2 with the N-terminal mutants to restore silencing and methylation. It would have been much better to generate constructs that express normal levels of these two proteins, because their data at present show no functional role for Suv39h1 nor its N-terminal domain in vivo.

2) With regards to 1, the data in Figure 2 show that Suv39h2 N term is not required for repression of MSR RNAs or for full levels of H3K9me3 (Figure 2). In these experiments, the authors introduce wild type or mutant Suv39h2 D(T3K81)-EGFP, into Suv39h1/h2 double knockout cells. So, the result cannot be explained by possible redundancy in RNA binding activity between Suv39h1 and h2. What then is the physiological significance of the described RNA binding activity and its contribution to stable retention of Suv39h2 at heterochromatin?

3) Data in Figure 5 nicely use a biochemical assay to show that Suv39h1 and h2 associate with polynucleosomes in an RNase-sensitive manner. One concern here is that the Suv39h2 N term mutant shows association with poly-nucleosomes that is similar to the WT protein. This complicates the conclusion of the assay regarding the role of the N term in RNA binding, since association is not affected by deletion of the domain that mediates RNA binding. The authors do see higher sensitivity of the N term mutant to RNase treatment at 350 mM salt and interpret this to mean a role for the N term in ssRNA binding (which is digested by RNase A even at 350 mM salt). This doesn't make sense, unless we are missing something. The result could also be interpreted to mean that the N term is involved in chromatin association independently of RNA. The mutant may be more sensitive to RNase digestion because another domain mediates the RNA-sensitive interaction. This needs to be clarified.

4) How does Suv39h1 achieve repression of major satellites in NZ arrested cells without it binding to mitotic chromosomes? (Figure 2).

5) It will be important for them to show that RNaseH /A treatment of cells leads to loss of Suv39h1 and 2 from heterochromatin by immunofluorescence.

---

## [Author Response]

*Summary:*

*The compaction of repetitive DNA elements into heterochromatin is critical for genome stability in eukaryotes. Heterochromatin is molecularly defined by the abundance of histone H3 lysine 9 trimethylation (H3K9me), to which several H3K9me-binding proteins bind, recruiting silencing activities. The Suv39 family of lysine methyltransferases (KMT) catalyze H3K9 methylation in eukaryotes. How Suv39h proteins are recruited to heterochromatin has been the subject of several studies in plants, the fission yeast. and germ cells of flies, worms and mammals. These studies have shown that small and long nuclear noncoding RNAs cooperate to recruit the Suv39 family proteins to repetitive DNA elements, but whether and how nuclear RNAs also play a role in the recruitment of Suv39 proteins in mammalian somatic cells remains unknown. The paper by Camacho et al. addresses this question in mice, demonstrating that the mouse Suv39h2 N-terminal region possesses an RNA binding domain, which can bind to single-stranded long heterochromatic transcripts (such as major satellite repeat (MSR) and TERRA) with high affinity in vitro. They also show that MSR RNAs are tethered to chromatin and that these chromatin-associated RNAs are required for the stable association of Suv39 KMTs to polynucleosomes. These experiments are generally well-controlled and redundantly support each other, and the data are of excellent quality. Unfortunately, these data indicate no functional role for the N-terminal RNA binding domain of Suv39h2 in vivo. However, there is a concern that potential overexpression of the reintroduced Suv39h1 and Suv39h2 may have obliterated the ability to observe a functional requirement for both Suv39h1 and Suv39h2 or to see the functional requirement for the N-terminus of Suv39h2 in vivo.*

We disagree that there would be no functional role for the basic domain of Suv39h2 in vivo. The novel data on confocal imaging of mitotic chromosomes (new Figure 2) and the previous data on chromatin retention with mitotically enriched chromatin (Figure 2) and RNaseA-mediated dissociation from poly-nucleosomes (Figure 5) all indicate a physiological role for the basic domain of Suv39h2 to ensure robust chromatin association for the full-length Suv39h2 enzyme that persists during mitosis. We have now clearly explained this in a new paragraph of the Discussion (subsection “MSR RNA govern stable association of Suv39h enzymes with chromatin”, second paragraph). In addition, although the analysis on the basic domain of Suv39h2 is a significant part of the study, another major part of our work provides detailed and novel insight into the chemical properties and biochemical functions of the MSR non-coding RNA that, we feel, is another important advance.

*Essential revisions:*

*1) How do the expression levels of the reintroduced GFP tagged Suv39h1 and h2 proteins compare to endogenous levels, given that they have more robust silencing and more H3 K9 methylation of major satellites in NZ arrested cells than the wild type (Figure 2)? It would seem likely that the levels may be a lot higher than endogenous levels given that they lead to a much greater silencing of transcription, while the basic domain deletion was reduced to wild type levels. Western blotting with an antibody against Suv39h1 and 2 needs to be included on these westerns. If the levels of the ectopic proteins are a lot higher than endogenous levels, the results need to be discussed and considered in light of this. It would be very unfortunate, because this could explain why either Suv39h1 (which lacks the exon including the N-terminal RNA binding domain) or Suv39h2 can complement the Suv39h1/2 double null mouse cells. This would also explain the ability of Suv39h2 with the N-terminal mutants to restore silencing and methylation. It would have been much better to generate constructs that express normal levels of these two proteins, because their data at present show no functional role for Suv39h1 nor its N-terminal domain in vivo.*

We have added the Western analysis for the endogenous Suv39h1 and Suv39h2 protein levels (new Figure 1). The data show that there is no massive overexpression of the reintroduced Suv39h-EGFP products and that full-length Suv39h2-EGFP is not dominating the protein expression (subsection “Suv39h1 and Suv39h2 can independently re-establish H3K9me3 and silence 147 MSR transcription in interphase mouse ES cells”, first paragraph).

*2) With regards to 1, the data in Figure 2 show that Suv39h2 N term is not required for repression of MSR RNAs or for full levels of H3K9me3 (Figure 2). In these experiments, the authors introduce wild type or mutant Suv39h2 D(T3K81)-EGFP, into Suv39h1/h2 double knockout cells. So, the result cannot be explained by possible redundancy in RNA binding activity between Suv39h1 and h2. What then is the physiological significance of the described RNA binding activity and its contribution to stable retention of Suv39h2 at heterochromatin?*

As indicated above, we have added a new paragraph in the Discussion (subsection “MSR RNA govern stable association of Suv39h enzymes with chromatin”, second paragraph) that clearly describes the physiological role of the basic domain of Suv39h2 in providing "an additional RNA affinity to the Suv39h2 enzyme, which then allows for more robust heterochromatin association of Suv39h2 that persists during mitosis".

*3) Data in Figure 5 nicely use a biochemical assay to show that Suv39h1 and h2 associate with polynucleosomes in an RNase-sensitive manner. One concern here is that the Suv39h2 N term mutant shows association with poly-nucleosomes that is similar to the WT protein. This complicates the conclusion of the assay regarding the role of the N term in RNA binding, since association is not affected by deletion of the domain that mediates RNA binding. The authors do see higher sensitivity of the N term mutant to RNase treatment at 350 mM salt and interpret this to mean a role for the N term in ssRNA binding (which is digested by RNase A even at 350 mM salt). This doesn't make sense, unless we are missing something. The result could also be interpreted to mean that the N term is involved in chromatin association independently of RNA. The mutant may be more sensitive to RNase digestion because another domain mediates the RNA-sensitive interaction. This needs to be clarified.*

This is a correct criticism and we have clarified this with a revised paragraph in the Discussion (subsection “MSR RNA govern stable association of Suv39h enzymes with chromatin”, first paragraph). We clearly state that several modules in the Suv39h proteins contribute to RNA interaction and also cite the independent work by the laboratories of Yoichi Shinkai and Aaron Straight that shows RNA binding by the chromo domain of mouse Suv39h1 and of human SUV39H1. In our work, we have focused on the definition of the basic domain of Suv39h2 as a novel RNA interaction motif that donates an additional RNA affinity to Suv39h2.

*4) How does Suv39h1 achieve repression of major satellites in NZ arrested cells without it binding to mitotic chromosomes? (Figure 2).*

Although there is transient dissociation of Suv39h1 from mitotic chromosomes, H3K9me3 persists during mitosis (Figure 2).

*5) It will be important for them to show that RNaseH /A treatment of cells leads to loss of Suv39h1 and 2 from heterochromatin by immunofluorescence.*

As already discussed in the original manuscript, we have done RNase treatment of permeabilized cells but have not observed dispersion of Suv39h enzymes from heterochromatic foci in interphase. We have revisited this and repeated RNaseA and RNaseH treatments under various salt and pH buffer conditions, but again have not seen dispersion from heterochromatic foci in interphase. Figure 8 documents these experiments. We have not done these experiments with mitotic chromosomes. We do know from informal exchange with Aaron Straight (independent manuscript) that human SUV39H1 is dispersed from mitotic chromosomes upon RNaseA treatment. As there is cell-cycle regulated phosphorylation of SUV39H1 (Aagard et al. 2000), mitotic (and phosphorylated) SUV39H1 may be more sensitive to RNaseA treatment as compared to interphase Suv39h1 or Suv39h2. We have clarified this in the Discussion (subsection “MSR RNA govern stable association of Suv39h enzymes with chromatin”, last paragraph).

Author response image 1.**DOI:**
http://dx.doi.org/10.7554/eLife.25293.019